# Adaptive laboratory evolution enhances methanol tolerance and conversion in engineered *Corynebacterium glutamicum*

Yu Wang [1], Liwen Fan[1,2], Philibert Tuyishime[1], Jiao Liu[1], Kun Zhang[1,3], Ning Gao[1,3], Zhihui Zhang[1,3], Xiaomeng Ni[1], Jinhui Feng[1], Qianqian Yuan[1], Hongwu Ma[1], Ping Zheng[1,2,3 ✉], Jibin Sun[1,3 ✉] & Yanhe Ma[1]

Synthetic methylotrophy has recently been intensively studied to achieve methanol-based biomanufacturing of fuels and chemicals. However, attempts to engineer platform micro-organisms to utilize methanol mainly focus on enzyme and pathway engineering. Herein, we enhanced methanol bioconversion of synthetic methylotrophs by improving cellular tolerance to methanol. A previously engineered methanol-dependent *Corynebacterium glutamicum* is subjected to adaptive laboratory evolution with elevated methanol content. Unexpectedly, the evolved strain not only tolerates higher concentrations of methanol but also shows improved growth and methanol utilization. Transcriptome analysis suggests increased methanol concentrations rebalance methylotrophic metabolism by down-regulating glycolysis and up-regulating amino acid biosynthesis, oxidative phosphorylation, ribosome biosynthesis, and parts of TCA cycle. Mutations in the *O*-acetyl-ʟ-homoserine sulfhydrylase Cgl0653 catalyzing formation of ʟ-methionine analog from methanol and methanol-induced membrane-bound transporter Cgl0833 are proven crucial for methanol tolerance. This study demonstrates the importance of tolerance engineering in developing superior synthetic methylotrophs.

[1] Key Laboratory of Systems Microbial Biotechnology, Tianjin Institute of Industrial Biotechnology, Chinese Academy of Sciences, Tianjin 300308, China. [2] School of Life Sciences, University of Science and Technology of China, Hefei 230026, China. [3] University of Chinese Academy of Sciences, Beijing 100049, China. ✉email: zheng_p@tib.cas.cn; sun_jb@tib.cas.cn

Biomanufacturing utilizes renewable feedstocks for production of fuels and chemicals that are traditionally produced from fossil fuel-based processes. While easily accessible sugars are still the dominating feedstocks for biomanufacturing, efforts to replace sugars with non-food and cheaper alternative feedstocks never cease[1]. The broadly available and energy-rich C1 liquid compound, methanol, has emerged as a promising candidate[2]. Methanol can be readily produced from methane, the major component of natural gas with an estimated worldwide recoverable amount of over 7 quadrillion ft[3]. It can also be produced renewably from municipal solid waste and residual biomass through syngas as an intermediate, and by electrocatalytic reduction or hydrogenation of greenhouse gas carbon dioxide[4]. Currently, the price of methanol is comparable to that of glucose, despite the fact that methanol can provide 50% more reducing power for cell growth and biosynthesis[5].

To implement methanol-based biomanufacturing, microbial cell factories capable of efficiently utilizing methanol as a carbon source are urgently needed[6]. Although native methylotrophs possess the ability to grow on methanol, there has been few success in employing them to produce valuable compounds at high levels[7,8]. Engineering of native methylotrophs is difficult and time-consuming due to lack of advanced genetic tools and metabolic knowledge. For these reasons, the concept of synthetic methylotrophy, which aims to integrate methanol assimilation into genetically tractable hosts for methanol bioconversion, has arisen and received increasing attention[3,5].

After initial attempts to engineer synthetic methylotrophy in *Escherichia coli* and *Corynebacterium glutamicum* by heterologous expression of NAD-dependent methanol dehydrogenase (Mdh) and ribulose monophosphate (RuMP) pathway enzymes 3-hexulose-6-phosphate synthase (Hps) and 6-phospho-3-hexuloisomerase (Phi)[9,10], massive efforts have been devoted to breaking through the bottlenecks to efficient methanol assimilation with the ultimate purpose of utilizing methanol as the sole carbon source[5]. Artificial pathways have also been developed for assimilation of C1 compounds including methanol[11–17]. Based on the current knowledge, the poor kinetics of NAD-dependent Mdh and insufficient supply of formaldehyde acceptor ribulose-5-phosphate (Ru5P) are two major bottlenecks. Discover and directed evolution of more active Mdh candidates have been conducted to address the first bottleneck[18–20]. The strategy of enzyme co-localization and metabolite channeling has also been applied to drive oxidation of methanol to formaldehyde by constructing Mdh-Hps-Phi complexes[21,22]. To regenerate Ru5P more efficiently, the non-oxidative pentose phosphate pathway (PPP) from native methylotroph *Bacillus methanolicus* or the sedoheptulose-bisphosphatase (SBPase) from *E. coli* was overexpressed in synthetic methylotrophs to activate the SBPase pathway variant of the RuMP cycle[23,24]. To further improve methanol assimilation, methanol-dependent synthetic methylotrophs were engineered by blocking the Ru5P catabolism, which forced cells to utilize methanol under a co-consumption regime with a Ru5P source (e.g. xylose, ribose, gluconate). Adaptive laboratory evolution (ALE) strategies, which allow occurrence and selection of beneficial mutations in an unbiased fashion[25], were then applied to effectively improve cell growth on methanol and co-substrates[26–28].

The effects of methanol tolerance on methanol utilization have long been neglected. In comparison to normally used feedstocks such as sugars and glycerol, methanol is more cytotoxic to cells. It is suggested that as a solvent, methanol would cause oxidative stress and affect the fluidity and mechanical stability of cellular membrane, leading to membrane disruption[29]. Therefore, methanol is usually used at a concentration lower than 250 mM (8 g/L) for representative native methylotrophs *B. methanolicus*

MGA3[30] and *Methylobacterium extorquens* AM1[31] and methylotrophic *E. coli*[13,15,18,23,26,32] and *C. glutamicum* strains[10,28,33], except that a methanol-essential *E. coli* strain could tolerate 500 mM methanol[27].

*C. glutamicum* is one of the most important industrial workhorses due to its GRAS status (generally regarded as safe), relatively few growth requirements, and ability to produce and secrete large amounts of amino acids[34]. We previously engineered and evolved *C. glutamicum* strain MX-11 by ALE for methanol-dependent growth and amino acid production[28]. In this study, strain MX-11 is sequentially evolved under increased methanol content and better mutants with improved methanol tolerance, growth rate, and methanol utilization are screened. Transcriptome and genome analyses are subsequently conducted to reveal the regulatory and genetic factors responsible for the improved performance. Crucial mutations are identified and studied for their physiological functions in resistance to high concentrations of methanol. This study provides new genetic targets for engineering synthetic methylotrophy and underscores the value of tolerance engineering in constructing a superior methanol utilizer.

## Results

**Adapting *C. glutamicum* to higher concentrations of methanol.** Our previously constructed and evolved methanol-dependent *C. glutamicum* MX-11 grew on methanol and xylose as co-substrates (Fig. 1a)[28]. However, the cells only tolerated methanol up to 4 g/L. Growth was seriously inhibited when methanol increased to 10 g/L or higher concentrations (Fig. 1b). In order to improve its tolerance to methanol, strain MX-11 was continuously cultivated in CGXII minimal medium supplemented with 15 g/L methanol and 4 g/L xylose. The specific growth rate of strain MX-11 gradually increased from $0.005\,h^{-1}$ to $0.016\,h^{-1}$ (a 3.20-fold increase) after seven passages of ALE (Fig. 1c). Then three evolved strains with largely improved cell growth on 15 g/L methanol were isolated and designated as MX-12, MX-13, and MX-14, respectively. The highest cell biomass of the evolved strains cultivated with 15 g/L methanol was 6.68-fold and 1.54-fold higher than the parent strain MX-11 cultivated with 15 g/L and 4 g/L methanol, respectively (Fig. 1d). In the absence of methanol, the evolved strains showed no growth with xylose as the sole carbon source, suggesting that ALE did not change the methanol-dependent feature and methanol was still an indispensable carbon source.

**Improving methanol tolerance enhances methanol conversion.** One of the screened mutants, strain MX-14, was further characterized by its growth and substrate uptake with different concentrations of methanol due to its slight growth advantage relative to the rest two mutants. With 4 g/L methanol and 4 g/L xylose, strain MX-14 showed a similar growth rate with its parent strain MX-11 under the same cultivation conditions (Figs. 1b and 2a). The growth advantage of strain MX-14 appeared with higher concentrations of methanol. Cultivation using 10 g/L, 15 g/L, and 20 g/L methanol resulted in 1.17-, 1.33-, and 1.24-fold increases in cell biomass compared to that obtained using 4 g/L methanol. The specific growth rates during exponential growth in 10 g/L, 15 g/L, and 20 g/L methanol reached $0.040\,h^{-1}$, $0.050\,h^{-1}$, and $0.052\,h^{-1}$, which were 1.38-, 1.71-, and 1.79-fold higher than that obtained in 4 g/L methanol, respectively (Fig. 2a). Higher concentrations of methanol also led to higher substrate uptake rates (Fig. 2b, c). The specific methanol uptake rates during exponential growth with higher concentrations of methanol reached 2.12–2.51 mmol/gCDW·h, which increased by over 2-fold compared to cells cultivated with 4 g/L methanol. The parent strain

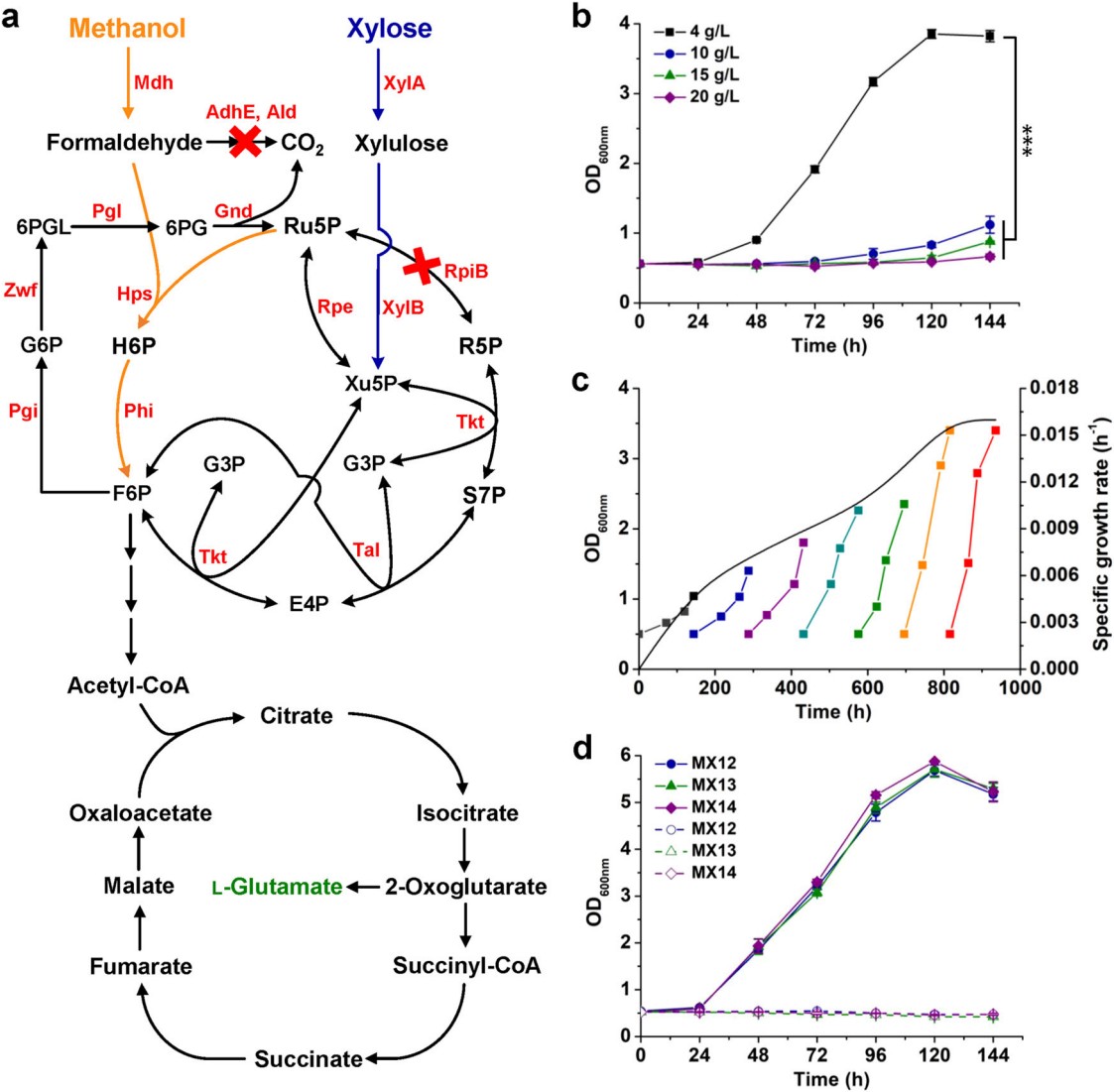

**Fig. 1 Improving the tolerance to methanol via ALE. a** Detailed enzymatic reactions and metabolic pathways of the methanol-dependent *C. glutamicum*. Enzymes: methanol dehydrogenase (Mdh), 3-hexulose-6-phosphate synthase (Hps), 6-phospho-3-hexuloisomerase (Phi), mycothiol-dependent formaldehyde dehydrogenase (AdhE), acetaldehyde dehydrogenase (Ald), xylose isomerase (XylA), xylulokinase (XylB), ribose phosphate isomerase (RpiB), ribulose phosphate epimerase (Rpe), transketolase (Tkt), transaldolase (Tal), glucose-6-phosphate isomerase; (Pgi), glucose-6-phosphate 1-dehydrogenase (Zwf), 6-phosphogluconolactonase (Pgl), 6-phosphogluconate dehydrogenase (Gnd). Metabolites: ribose-5-phosphate (R5P), ribulose-5-phosphate (Ru5P), xylulose-5-phosphate (Xu5P), glyceraldehyde-3-phosphate (G3P), erythrose-4-phosphate (E4P), sedoheptulose-7-phosphate (S7P), fructose-6-phosphate (F6P), hexulose-6-phosphate (H6P), glucose-6-phosphate (G6P), 6-phospho-glucono-1,5-lactone (6PGL), and 6-phospho-gluconate (6PG). **b** Effects of methanol concentration on cell growth of methanol-dependent *C. glutamicum* MX-11. ***$P < 0.001$, one-way ANOVA, $N = 3$, $P = 7.1 \times 10^{-11}$. **c** ALE of strain MX-11 in CGXII minimal medium supplemented with 15 g/L methanol and 4 g/L xylose. The specific growth rate for each passage of ALE (the black curve) was calculated using the $OD_{600nm}$ values at the initial and final time points. **d** Growth curve of evolved methanol-dependent strains. Strains were cultivated using CGXII minimal medium supplemented with 15 g/L methanol and 4 g/L xylose (solid lines) or only xylose (dotted lines) as the carbon source(s). Values and error bars reflect the mean ± s.d. of three biological replicates ($N = 3$).

MX-11 co-utilized methanol and xylose with an average mole ratio of 3.83:1[28]. During cultivation with 15 g/L methanol, the evolved strain MX-14 consumed 6.52 g/L methanol (203.75 mM) and 4.34 g/L xylose (28.93 mM). The co-utilization ratio of methanol ($C_1$) and xylose ($C_5$) reached 7.04:1, demonstrating that methanol was the major carbon source. It was noticed that cell growth ceased at ~120 h when xylose was exhausted but methanol was not (Fig. 2). Deactivation of ribose phosphate isomerase (RpiB) coupled cell growth with methanol and xylose co-utilization. However, when xylose was exhausted, cells cannot maintain growth with methanol as the sole carbon source.

Improved tolerance to methanol allowed strain MX-14 to grow with higher concentrations of methanol, and therefore utilize methanol more efficiently. To test whether methanol-based bioproduction was also enhanced, strain MX-14 was applied to produce L-glutamate, the largest product segment within the amino acid market. Penicillin G was added to induce L-glutamate production of strain MX-14 at the middle exponential phase as described previously for the parent strain MX-11[28]. After the inducing treatment, cell growth was inhibited and L-glutamate production was initiated. After 144 h of fermentation, 230 mg/L extracellular L-glutamate was produced (Fig. 2d). The titer was

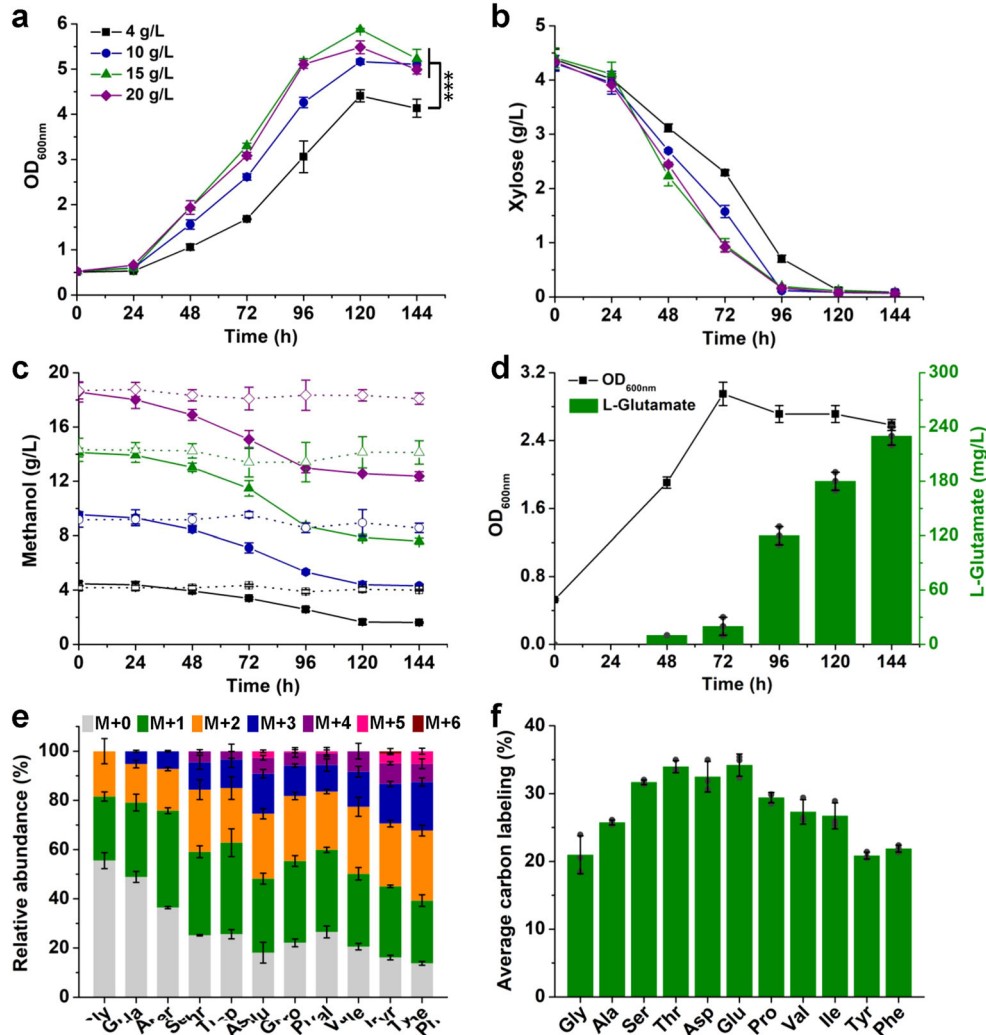

**Fig. 2 Enhanced methanol bioconversion by evolved methanol-dependent *C. glutamicum* MX-14. a–c** Growth curve (**a**), xylose utilization (**b**), and methanol utilization (**c**) of strain MX-14 with different concentrations of methanol. Strain MX-14 was cultivated using CGXII minimal medium supplemented with 4, 10, 15, or 20 g/L methanol and 4 g/L xylose as the carbon sources. An evaporation control without inoculation of strain MX-14 was conducted simultaneously (dotted lines in **c**). ***$P < 0.001$, one-way ANOVA, $N = 3$, $P = 1.2 \times 10^{-4}$. **d** L-Glutamate production from methanol and xylose by strain MX-14. *C. glutamicum* MX-14 was cultivated in CGXII minimal medium supplemented with 15 g/L methanol and 4 g/L xylose. To induce L-glutamate production, penicillin G was added to a final concentration of 60 U/mL when the OD$_{600nm}$ of the culture reached ~3.0. **e** Relative abundance of proteinogenic amino acid mass isotopomers. **f** Average $^{13}$C-labeling of proteinogenic amino acids. *C. glutamicum* MX-14 was cultivated in CGXII minimal medium supplemented with 15 g/L $^{13}$C-methanol and 4 g/L non-labeled xylose. Cells were collected at 120 h for $^{13}$C-labeling analysis. Values and error bars reflect the mean ± s.d. of three biological replicates ($N = 3$).

2.56-fold higher than that produced by the parent strain MX-11 with 4 g/L methanol[28].

$^{13}$C-methanol labeling approach has been applied to measure methanol incorporation into cellular biomass[35]. To further demonstrate the enhanced methanol assimilation, biomass samples of strain MX-14 cultivated with 15 g/L $^{13}$C-methanol and 4 g/L non-labeled xylose were collected, hydrolyzed and analyzed for $^{13}$C-labeling in proteinogenic amino acids using GC/Q-TOF-MS. All the detected amino acids were $^{13}$C-labeled, including completely labeled L-glycine, L-alanine, L-serine, L-threonine, L-aspartate, L-glutamate, L-proline, and L-valine (Fig. 2e). Production of these multiple-carbon labeled amino acids suggest that the formaldehyde accepter is partially provided through the cycling RuMP pathway or oxidative PPP, not only from exogenous xylose. The results were consistent with the exceeded equimolar consumption of methanol and xylose mentioned above. The average carbon labeling levels of these amino acids were between 20% and 30% (Fig. 2f), which were

1.20- to 1.70-fold higher than those of the parent strain MX-11 with 4 g/L methanol[28], indicating more methanol was assimilated into biomass. Taken together, the results demonstrate that improving methanol tolerance is an effective strategy to enhance methanol bioconversion.

**Transcriptome analysis reveals metabolic regulation.** To investigate the mechanism of improved methanol bioconversion under higher methanol concentrations, transcriptome analysis of strain MX-14 cultivated with 15 g/L or 4 g/L methanol was conducted. The same amount of xylose (4 g/L) was added as a co-substrate for two cultures. Pearson's correlation coefficient test and principle component analysis (PCA) indicate the good accuracy and repeatability of the experimental methods (Supplementary Fig. 1a, b). Transcript levels of total 452 genes were significantly changed upon altered methanol concentration. When cells were cultivated with 15 g/L methanol, 310 and 142

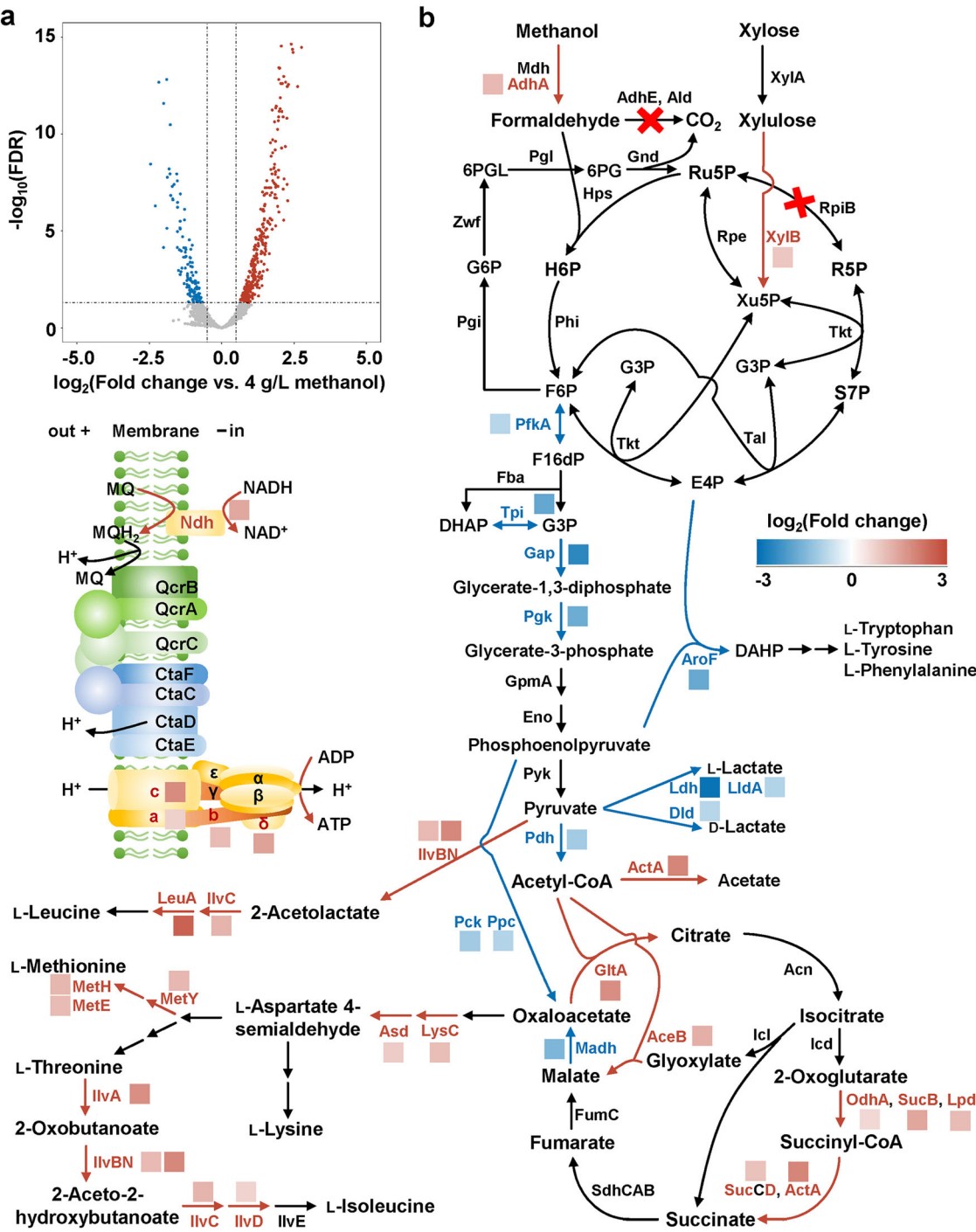

genes were up-regulated and down-regulated, respectively, relative to cultivation with 4 g/L methanol (Fig. 3a and Supplementary Data 1). These genes were classified into 29 cellular processes according to KEGG_B_class annotation, including amino acid metabolism, carbohydrate metabolism, energy metabolism, etc. (Supplementary Fig. 1c). Detailed analyses of the central metabolic pathway suggest down-regulation of glycolysis and pyruvate metabolism, which seems contrary to the improved cell growth under higher concentrations of methanol (Fig. 3b). However, a previous study has demonstrated the blockage of glycolysis benefits methanol assimilation in synthetic methylotrophs by driving carbon flux to regeneration of formaldehyde acceptor Ru5P, which is insufficient to sustain methanol assimilation[24]. The

down-regulated glycolysis, in association with the up-regulated alcohol dehydrogenase AdhA that functions as a native Mdh[36] and xylulokinase XylB that is involved in xylose metabolism, is expected to facilitate efficient methanol uptake and assimilation. Interestingly, part of the TCA cycle was enhanced but the conversion of malate to oxaloacetate by NAD-dependent malate dehydrogenase (Madh) was attenuated (Fig. 3b). It is coincidentally consistent with a previous study that demonstrates deactivation of Madh in *E. coli* is metabolically beneficial for synthetic methylotrophy, possibly by influencing the $NAD^+$/NADH ratio[27]. These considerations are supported by the fact that some methylotrophic bacteria have a low TCA cycle activity or even an incomplete TCA cycle[37,38].

**Fig. 3 Transcriptome analysis of *C. glutamicum* strain MX-14 cultivated with CGXII minimal medium supplemented with 15 g/L or 4 g/L methanol and 4 g/L xylose. a** Volcano plot of differential transcript levels as determined by RNA sequencing ($n = 3$). **b** Gene transcript level changes related to central metabolism pathways and oxidative phosphorylation between MX-14 cultivated with 15 g/L vs. 4 g/L methanol. Significant differentially expressed genes were defined as having a false discovery rate (FDR) < 0.05 and a $\log_2$(Fold change) >0.5 or <−0.5. Up-regulated and down-regulated enzymes are indicated with red and blue, respectively. Enzymes: alcohol dehydrogenase (AdhA), xylulokinase (XylB), 6-phosphofructokinase (PfkA), fructose bisphosphate aldolase (Fba), triosephosphate isomerase (Tpi), glyceraldehyde-3-phosphate dehydrogenase (Gap), phosphoglycerate kinase (Pgk), phosphoglycerate mutase (GpmA), enolase (Eno), pyruvate kinase (Pyk), FMN-dependent L-lactate dehydrogenase (LldA), NAD-dependent L-lactate dehydrogenase (Ldh), FAD/FMN-containing D-lactate dehydrogenase (Dld), pyruvate dehydrogenase (Pdh), acetyl-CoA hydrolase (ActA), phosphotransacetylase (Pta), phosphoenolpyruvate carboxykinase (Pck), phosphoenolpyruvate carboxylase (Ppc), citrate synthase (GltA), aconitate hydratase (Acn), isocitrate dehydrogenase (Icd), α-oxoglutarate dehydrogenases E1 component (OdhA), dihydrolipoamide acyltransferases (SucB), dihydrolipoamide dehydrogenase (Lpd), succinyl-CoA synthetase (SucCD), succinate dehydrogenase (SdhCAB), fumarate hydratase (FumC), malate dehydrogenase (Madh), isocitrate lyase (Icl), malate synthase (AceB), 3-deoxy-7-phosphoheptulonate synthase (AroF), aspartokinase (LysC), aspartate-semialdehyde dehydrogenase (Asd), homoserine kinase (ThrB), O-acetyl-L-homoserine sulfhydrylase (MetY), methionine synthase I (MetH), methionine synthase II (MetE), threonine dehydratase (IlvA), acetolactate synthase (IlvBN), ketol-acid reductoisomerase (IlvC), dihydroxyacid dehydratase (IlvD), branched-chain amino acid aminotransferase (IlvE), isopropylmalate synthase (LeuA), FAD-containing NADH dehydrogenase (Ndh), cytochrome $bc_1c$ complex (QcrCAB), cytochrome $aa_3$ complex (CtaCDEF), ATP synthase (α, Cgl1210; β, Cgl1212; γ, Cgl1211, δ, Cgl1209; ε, Cgl1213; a, Cgl1206; b, Cgl1208; c, Cgl1207). Metabolites: fructose-1,6-bisphosphate (F16dP), dihydroxyacetone phosphate (DHAP), 3-deoxy-arabino-heptulonate 7-phosphate (DAHP), menaquinone (MQ), and menaquinol (MQH$_2$).

Accelerated cell growth will require an increased supply of organic skeleton material and energy. Many genes involved in metabolism of amino acid, nucleotide, and energy are up-regulated upon increased methanol content (Supplementary Fig. 1c). Specifically, biosynthesis of L-aspartate family amino acids (L-lysine, L-methionine, L-threonine, and L-isoleucine) and L-leucine was overall enhanced (Fig. 3b). More energy would be provided due to the up-regulation of partial respiratory chain, including NADH dehydrogenase that transfers electrons from NADH to menaquinone and several subunits of ATP synthase that synthesizes ATP using a proton gradient (Fig. 3b). The up-regulated NADH dehydrogenase might also lead to an elevated NAD$^+$/NADH ratio in favor of methanol utilization by thermodynamically supporting the oxidation of methanol. From another perspective, cell growth rate is suggested to be linearly correlated with the abundance of ribosomes that polymerize amino acids into proteins[39]. Indeed, protein translation-associated genes including ribosome encoding genes were significantly upregulated under a high concentration of methanol (Supplementary Fig. 1c and Supplementary Data 1). However, the increased ribosomal protein fraction would reduce the metabolic protein fraction[40]. Since the glycolytic enzymes account for a major fraction of cellular total proteins, down-regulation of glycolysis is supposed to save resources for ribosomal proteins synthesis, which may redistribute protein synthesis and support a faster cell growth rate[41].

**Genome sequencing of evolved methanol-tolerant strains**. To identify the genetic mutations responsible for the improved methanol tolerance and bioconversion especially in the presence of high concentrations of methanol, the genomes of evolved strains MX-12, MX-13, and MX-14 were sequenced. All the three evolved strains harbored 10 mutations, nine of which were inherited from their parent strain MX-11. Interestingly, each evolved strain accumulated only one more mutation during this round of ALE and all the mutated genes encoded membrane-bound proteins (Table 1). The additional mutation of strain MX-12 existed in gene *cgl2365* (nucleotide change C542G, amino acid change A181G), which was predicted to be a membrane protein with three transmembrane (TM) helices using TMHMM method[42]. The A181G mutation locates in the C-terminal of Cgl2365 and outside the TM helices. Unfortunately, *cgl2365* has neither been investigated for its biological function nor linked with a metabolic process, making interpretation of its function in methanol tolerance or bioconversion difficult. A synonymous

mutation was found in gene *cgl2857* (nucleotide change G183A) of strain MX-13. Cgl2857 is a membrane protein, that is highly conserved in Corynebacterineae and required for synthesis of full-length lipomannans and lipoarabinomannans, abundant components of the multilaminate cell wall[43]. *C. glutamicum* synthesizes a complex cell wall, which can confer intrinsic resistance to adverse environmental conditions[44]. Although synonymous mutations do not alter the encoded protein, they can influence gene expression[45]. Therefore, this mutation may affect the expression of Cgl2857, alter the cell wall structure, and consequently improve methanol tolerance. Strain MX-14 had a missense mutation in gene *cgl0833* (nucleotide change C1439T, amino acid change S480F). Cgl0833 is a monocarboxylic acid transporter (MctC) for uptake of pyruvate, acetate, and propionate. Its transcription is under control of global transcriptional regulators RamA and RamB[46]. It is unknown how this transporter is involved in methanol tolerance or metabolism.

**Identifying mutations responsible for methanol tolerance**. Since strains MX-12, MX-13, and MX-14 have only one more mutation than their parent strain MX-11 but grow well in 15 g/L methanol, we speculate that the three mutations are beneficial for methanol tolerance and bioconversion. The other mutations inherited from the parent strain MX-11 also have not been characterized for their roles in methanol utilization or tolerance. To test the functions of these mutations, MX-10, the parent strain of MX-11, was selected as a host. MX-10 harbors the four mutations in *cgl0111*, *cgl2030*, *cgl2192*, and *cgl2424* that accumulated during a pre-ALE using xylose and ribose as co-substrates[28]. Therefore, only the rest eight mutations were individually introduced to MX-10. Acquisition of these single-site mutations by strain MX-10 did not allow fast growth in CGXII minimal medium supplemented with methanol and xylose, suggesting the mutations collectively improved methanol-dependent growth. In LB medium with methanol and xylose, six of the eight tested mutations (*cgl0653*, *cgl0833*, *cgl1367*, *cgl1520*, *cgl2365*, and *cgl2857*) improved the growth of strain MX-10. The rest two mutations caused no positive or negative effect (Fig. 4a).

To better identify the mutations responsible for methanol tolerance, the single-site mutations were introduced to the wild-type *C. glutamicum* ATCC 13032 and the derivatives were tested for their growth in CGXII minimal medium supplemented with 5 g/L glucose as a carbon source and 30 g/L methanol as a stress condition. In the absence of methanol, the wild-type strain and its derivatives showed no difference in growth, suggesting that the

**Table 1 Mutations of evolved strains identified by genome sequencing.**

| Gene ID | Gene name | Gene product | Nucleotide alteration | Amino acid alteration | Strain |
|---|---|---|---|---|---|
| cgl0111 | atlR | Multi-function regulator of carbohydrate metabolism | T437G | I146S | MX-10, MX-11, MX-12, |
| cgl2030 | – | Predicted ATPase with chaperone activity | C535T | P179S | MX-13, and MX-14 |
| cgl2192 | ctaE | Cytochrome c oxidase subunit III | A433G | T145A | |
| cgl2424 | – | Uncharacterized membrane protein | C446A | A149E | |
| cgl0653 | metY | O-Acetyl-L-homoserine sulfhydrylase | G1256A | G419D | MX-11, MX-12, MX-13, |
| cgl0754 | mtrA | Dual regulator of genes involved in cell morphology, antibiotics susceptibility and osmoprotection | C582A | H194Q | and MX-14 |
| cgl1367 | uriR | Transcriptional repressor of uridine utilization and ribose uptake genes | C584T | T195I | |
| cgl1520 | – | Hypothetical protein | A574G | M192V | |
| cgl2998 | – | Hypothetical protein | G104T | G35V | |
| cgl2365 | – | Hypothetical membrane protein | C542G | A181G | MX-12 |
| cgl2857 | – | Membrane protein required for lipomannan maturation and lipoarabinomannan synthesis | G183A | T61T | MX-13 |
| cgl0833 | mctC | Monocarboxylic acid transporter | C1439T | S480F | MX-14 |

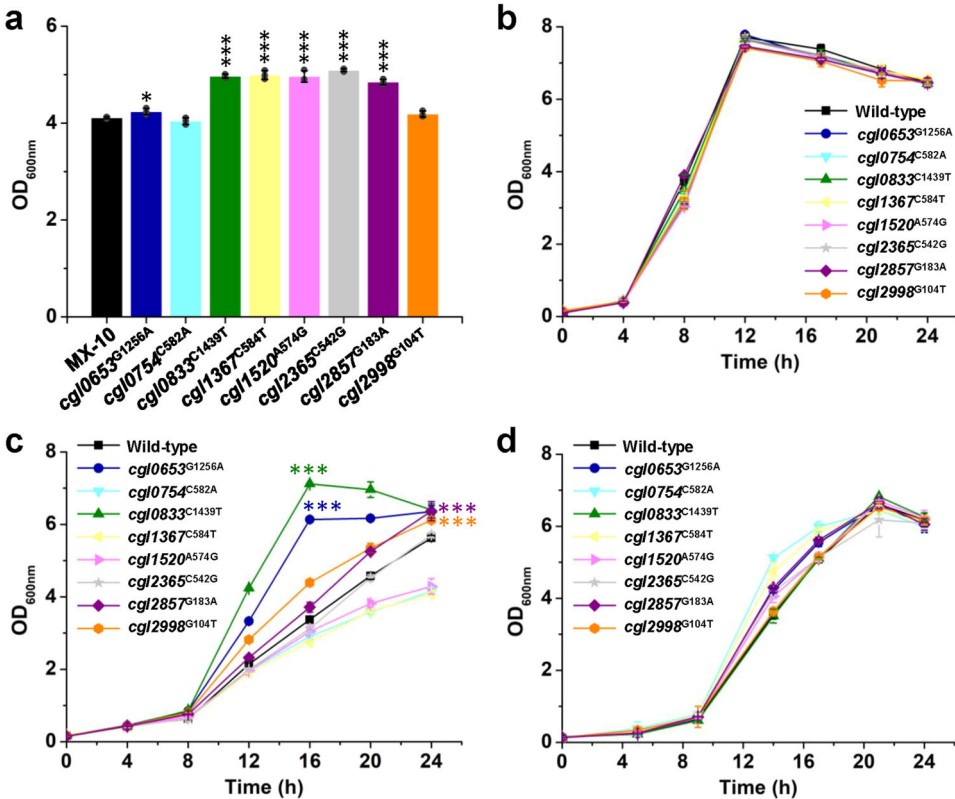

**Fig. 4 Effects of single-site mutations on methanol-dependent growth and methanol tolerance. a** Effects of single-site mutations on methanol-dependent growth of strain MX-10. Strain MX-10 and its derivatives harboring single-site mutations were cultivated using LB medium supplemented with 4 g/L methanol and 4 g/L xylose as the carbon sources. $OD_{600nm}$ values after 72 h cultivation were shown. $*P < 0.05$, $***P < 0.001$, one-way ANOVA, $N = 3$; $cgl0653^{G1256A}$, $P = 0.033$; $cgl0754^{C582A}$, $P = 0.17$; $cgl0833^{C1439T}$, $P = 5.1 \times 10^{-6}$; $cgl1367^{C584T}$, $P = 8.1 \times 10^{-5}$; $cgl1520^{A574G}$, $P = 2.5 \times 10^{-4}$; $cgl2365^{C542G}$, $P = 1.3 \times 10^{-6}$; $cgl2857^{G183A}$, $P = 2.8 \times 10^{-5}$; $cgl2998^{G104T}$, $P = 0.11$. **b–d** Growth curve of C. glutamicum ATCC 13032 wild-type strain and its derivatives harboring single-site mutations in CGXII minimal medium supplemented with 5 g/L glucose (**b**), 5 g/L glucose and 30 g/L methanol (**c**), and 5 g/L glucose and 15 mg/L formaldehyde (**d**). $***P < 0.001$, one-way ANOVA, $N = 3$; $cgl0653^{G1256A}$ vs. wild-type, $P = 2.4 \times 10^{-5}$; $cgl0833^{C1439T}$ vs. wild-type, $P = 1.9 \times 10^{-6}$; $cgl2857^{G183A}$ vs. wild-type, $P = 3.8 \times 10^{-5}$; $cgl2998^{G104T}$ vs. wild-type, $P = 4.0 \times 10^{-4}$. Values and error bars reflect the mean ± s.d. of three biological replicates ($N = 3$).

mutations caused no negative effect to cells (Fig. 4b). Under methanol stress, growth of all strains was hindered. However, the strains harboring $cgl0653^{G1256A}$ (originated from strain MX-11), $cgl2998^{G104T}$ (originated from strain MX-11), $cgl0833^{C1439T}$ (originated from strain MX-14), or $cgl2857^{G183A}$ (originated from strain MX-13) showed improved resistance (Fig. 4c). Above all, $cgl0653^{G1256A}$ and $cgl0833^{C1439T}$ largely improved growth under methanol stress. The specific growth rates during exponential growth of strains harboring $cgl0653^{G1256A}$ and $cgl0833^{C1439T}$ were 1.18-fold and 1.26-fold higher than the

wild-type strain, respectively. The cell biomass of strains harboring $cgl0653^{G1256A}$ and $cgl0833^{C1439T}$ after 16 h of cultivation was 1.82-fold and 2.11-fold higher than the wild-type strain, respectively (Fig. 4c). Because methanol can be oxidized to toxic intermediate formaldehyde by the native alcohol dehydrogenase AdhA[36], it is possible that the mutations indirectly improve methanol tolerance via improving cellular resistance to formaldehyde. To test this hypothesis, 15 mg/L formaldehyde stress was exerted on cells and hindered growth was observed. However, no mutant showed significant growth advantage compared to the wild-type strain, suggesting a direct association between the mutations and methanol tolerance (Fig. 4d). Because the $cgl0653^{G1256A}$ and $cgl0833^{C1439T}$ mutations had the biggest impact on methanol tolerance, their functions were further investigated.

**Cgl0653 mutation decreases L-methionine analog formation.** The $cgl0653$ gene encodes $O$-acetyl-L-homoserine sulfhydrylase (MetY) that catalyzes the conversion of $O$-acetyl-L-homoserine and sulfide to homocysteine, a precursor of L-methionine biosynthesis, and the direct biosynthesis of L-methionine from $O$-acetyl-L-homoserine and methanethiol[47]. In the presence of methanol, $O$-methyl-L-homoserine, an analog of L-methionine, is synthesized (Fig. 5a), which would cause growth defeat by producing dysfunctional proteins[48,49]. The $cgl0653^{G1256A}$ mutation leads to a L-glycine to L-aspartate substitution at the position 419 (G419D). We hypothesize that this mutation contributes to methanol tolerance by inhibiting the enzymatic side reaction of Cgl0653 and decreasing $O$-methyl-L-homoserine formation. To verify our hypothesis, the wild-type and mutant Cgl0653 proteins were heterogeneously expressed in $E.\ coli$ and purified. Interestingly, unlike the wild-type Cgl0653, Cgl0653$^{G419D}$ mostly existed in the form of inclusion body (Fig. 5b and Supplementary Fig. 2). Because $O$-acetyl-L-homoserine was not commercially available, a cascade reaction was set up to detect the $O$-methyl-L-homoserine formation activities of purified proteins with the help of L-homoserine acetyltransferase (MetX) (Fig. 5a). Liquid chromatography coupled with tandem mass spectrometry (LC-MS/MS) was used to measure the $O$-methyl-L-homoserine produced by the cascade reaction. Cgl0653$^{G419D}$ catalyzed the conversion of $O$-acetyl-L-homoserine and methanol to $O$-methyl-L-homoserine with a rate ~100-fold lower than that of the wild-type Cgl0653 (Fig. 5c). To better understand why the evolved mutation decreased activity, a homology model of $C.\ glutamicum$ Cgl0653 was constructed with a close homolog. However, the G419 is predicted to localize to the region far from the binding pocket of co-factor pyridoxal-5-phosphate and substrates methanol and $O$-acetyl-L-homoserine, which unlikely directly affects binding of co-factor or substrates (Supplementary Fig. 3).

The results suggest the G419D mutation negatively affects the structure and the catalytic activity of Cgl0653, leading to decreased $O$-methyl-L-homoserine formation and consequently improved methanol tolerance. Therefore, regulation of the Cgl0653 expression level should result in changes in cellular tolerance to methanol. To test this hypothesis, the wild-type $cgl0653$ and $cgl0653^{G1256A}$ were overexpressed via plasmid in the wild-type $C.\ glutamicum$ ATCC 13032. In the absence of methanol, no difference in cell growth between the strain overexpressing $cgl0653$ or $cgl0653^{G1256A}$ and the control harboring an empty plasmid. When strains were cultivated under the stress of 30 g/L methanol, growth defeat was observed, especially for the $cgl0653$-overexpressed strain. Since $cgl0653^{G1256A}$ encodes an enzyme variant with largely decreased activity towards $O$-methyl-L-homoserine formation, its overexpression caused only slight inhibition on late exponential growth (Fig. 5d). To repress the expression of $cgl0653$, CRISPR interference (CRISPRi) technology was applied by expressing the deactivated Cas9 (dCas9) and a guide RNA (gRNA) targeting $cgl0653$. A gRNA without the 20 nucleotide (nt) target-specific complementary region was used as the negative control. Similar with $cgl0653$ overexpression, knock-down of $cgl0653$ did not significantly affect growth without methanol stress. On the contrary, in the presence of 30 g/L methanol stress, knock-down of $cgl0653$ obviously improved growth in relative to the control strain (Fig. 5e). We further knocked out $cgl0653$ in the wild-type $C.\ glutamicum$ ATCC 13032 and observed a similar growth advantage in the presence of methanol stress (Fig. 5f). The results further suggest $O$-methyl-L-homoserine formation is an important mechanism of methanol toxicity and repression of $cgl0653$ is effective to improve cellular tolerance to methanol.

**Cgl0833 is down-regulated by mutation but induced by methanol.** $C.\ glutamicum$ is able to utilize a number of carbon sources, including sugars, organic acids, and alcohols. $cgl0833$ encodes a monocarboxylic acid transporter (MctC) that is responsible for uptake of monocarboxylic acids including pyruvate, acetate, and propionate[46]. To test whether the $cgl0833^{C1439T}$ mutation affects its biological function, growth assay using pyruvate, acetate, or propionate as the sole carbon source was conducted. Unexpectedly, the wild-type strain and $cgl0833^{C1439T}$ mutant strain showed similar growth on these organic acids (Fig. 6a). According to the annotation of KEGG, $cgl0833$ encodes a Na$^+$/proline, Na$^+$/panthothenate symporter. However, peptide uptake and amino acid export assay suggest the $cgl0833^{C1439T}$ mutation did not affect proline transport (Supplementary Fig. 4). It seems that $cgl0833$ possesses cryptic physiological functions that might be affected by the $cgl0833^{C1439T}$ mutation.

Occasionally, nucleotide changes influence not only the activities of encoding proteins but also the expression levels. To investigate the effect of $cgl0833^{C1439T}$ mutation on mRNA translation, a $gfp$ gene encoding green fluorescent protein (GFP) was fused to the C-terminal of wild-type $cgl0833$ or mutant $cgl0833^{C1439T}$ in the chromosome. This modification did not change the cell growth and the mutant expressing $cgl0833^{C1439T}$-$gfp$ fusion showed similar methanol tolerance to that expressing $cgl0833^{C1439T}$ (Fig. 6b). By detecting the GFP fluorescence, it was found that $cgl0833^{C1439T}$ reduced the expression of $cgl0833$ by 1.67-fold. Interestingly, methanol addition led to 2.23- and 1.29-fold increases in expression of $cgl0833$ and $cgl0833^{C1439T}$, respectively, suggesting the induction of $cgl0833$ by methanol (Fig. 6b). Since methanol will be oxidized to formaldehyde and formate through the native methanol dissimilation pathway in wild-type $C.\ glutamicum$ ATCC 13032[36], there is a possibility of formaldehyde or formate being the inducer. However, strains expressing $cgl0833$-$gfp$ showed enhanced GFP fluorescence only in the presence of methanol but not formaldehyde or formate, suggesting $cgl0833$ is specifically responded to methanol (Fig. 6c). To test whether $cgl0833$ was induced by methanol at transcription level, quantitative PCR (qPCR) was carried out. Indeed, methanol addition increased mRNA level of $cgl0833$ by 4.70-fold, whereas formaldehyde or formate did not cause induction effects (Supplementary Fig. 5), which was consistent with the GFP fluorescence assay.

Next, we investigated the effects of regulating $cgl0833$ expression on methanol tolerance. Gene overexpression in plasmid and gene repression via CRISPRi were conducted as previously described for regulation of $cgl0653$. Overexpression of

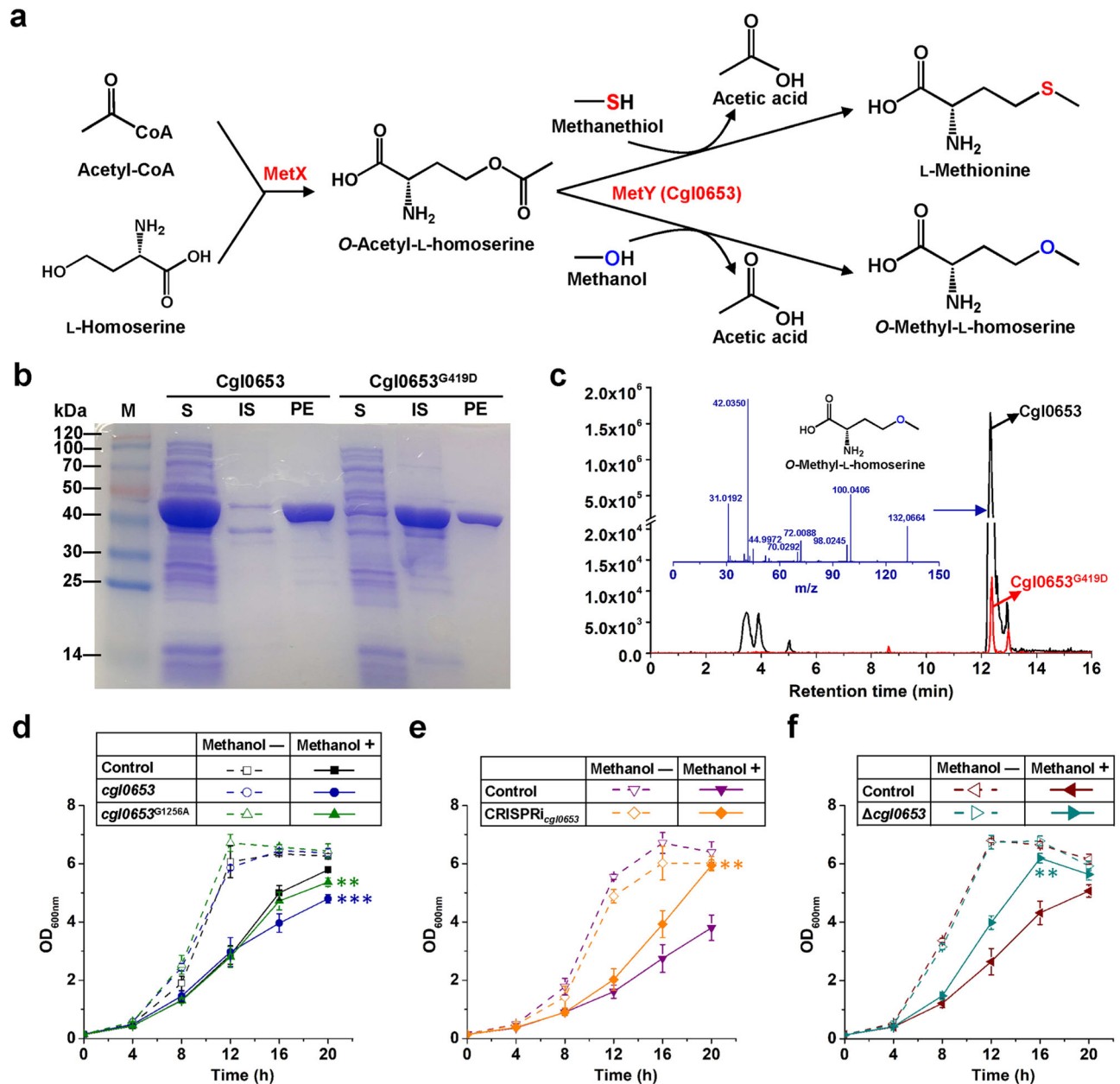

**Fig. 5 Effects of *cgl0653*<sup>G1256A</sup> mutation (amino acid change of G419D) on enzyme activity and methanol tolerance. a** Cascade reaction for Cgl0653 activity assay and *O*-methyl-ʟ-homoserine formation. ʟ-Homoserine acetyltransferase (MetX), O-Acetyl-ʟ-homoserine sulfhydrylase (MetY, Cgl0653). **b** Heterogeneous expression and purification of Cgl0653 and Cgl0653<sup>G419D</sup>. S Soluble supernatant of cell extract, IS insoluble sediment, PE purified enzyme. **c** LC-MS/MS analysis of *O*-methyl-ʟ-homoserine after 20 min reaction. **d** Effects of *cgl0653* and *cgl0653*<sup>G1256A</sup> overexpression on methanol tolerance. \*\**P* < 0.01, \*\*\**P* < 0.001, one-way ANOVA, *N* = 3; *cgl0653*, methanol + vs. control, methanol+, *P* = 3.0 × 10<sup>−4</sup>; *cgl0653*<sup>G1256A</sup>, methanol + vs. control, methanol+, *P* = 0.0070. **e** Effects of *cgl0653* knock-down on methanol tolerance. \*\**P* < 0.01, one-way ANOVA, *N* = 3; CRISPRi<sub>cgl0653</sub>, methanol + vs. control, methanol+, *P* = 0.0015. **f** Effects of *cgl0653* knock-out on methanol tolerance. \*\**P* < 0.01, one-way ANOVA, *N* = 3; Δ*cgl0653*, methanol + vs. control, methanol+, *P* = 0.0017. CGXII minimal medium supplemented with 5 g/L glucose and 30 g/L methanol was used to cultivate *C. glutamicum* ATCC 13032 and derivatives. IPTG (0.1 mM) was added at 4 h to induce *cgl0653* or *cgl0653*<sup>G1256A</sup> overexpression and dCas9 expression. Values and error bars reflect the mean ± s.d. of three biological replicates (*N* = 3).

either *cgl0833* or *cgl0833*<sup>C1439T</sup> could not improve the methanol tolerance of *C. glutamicum* ATCC 13032 (Fig. 6d). Conversely, knocking down *cgl0833* showed a positive effect on cell growth under the stress of 30 g/L methanol (Fig. 6e). Knock-out of *cgl0833* led to a similar growth advantage in the presence of methanol stress (Fig. 6f). Such results are reasonable considering the *cgl0833*<sup>C1439T</sup> mutation decreased the expression level of *cgl0833*. Although the mechanism of increased methanol tolerance by *cgl0833*<sup>C1439T</sup> mutation or *cgl0833* repression is still

unclear, *cgl0833* is a promising target for engineering cellular tolerance to methanol.

## Discussion

Considering the promising future of methanol-based biomanufacturing, great efforts have been devoted to developing synthetic methylotrophs for bioconversion of methanol to fuels and chemicals[3,7]. Design of artificial pathways[11–17], discovery and engineering of rate-limiting enzymes[18–20], and optimization of Ru5P

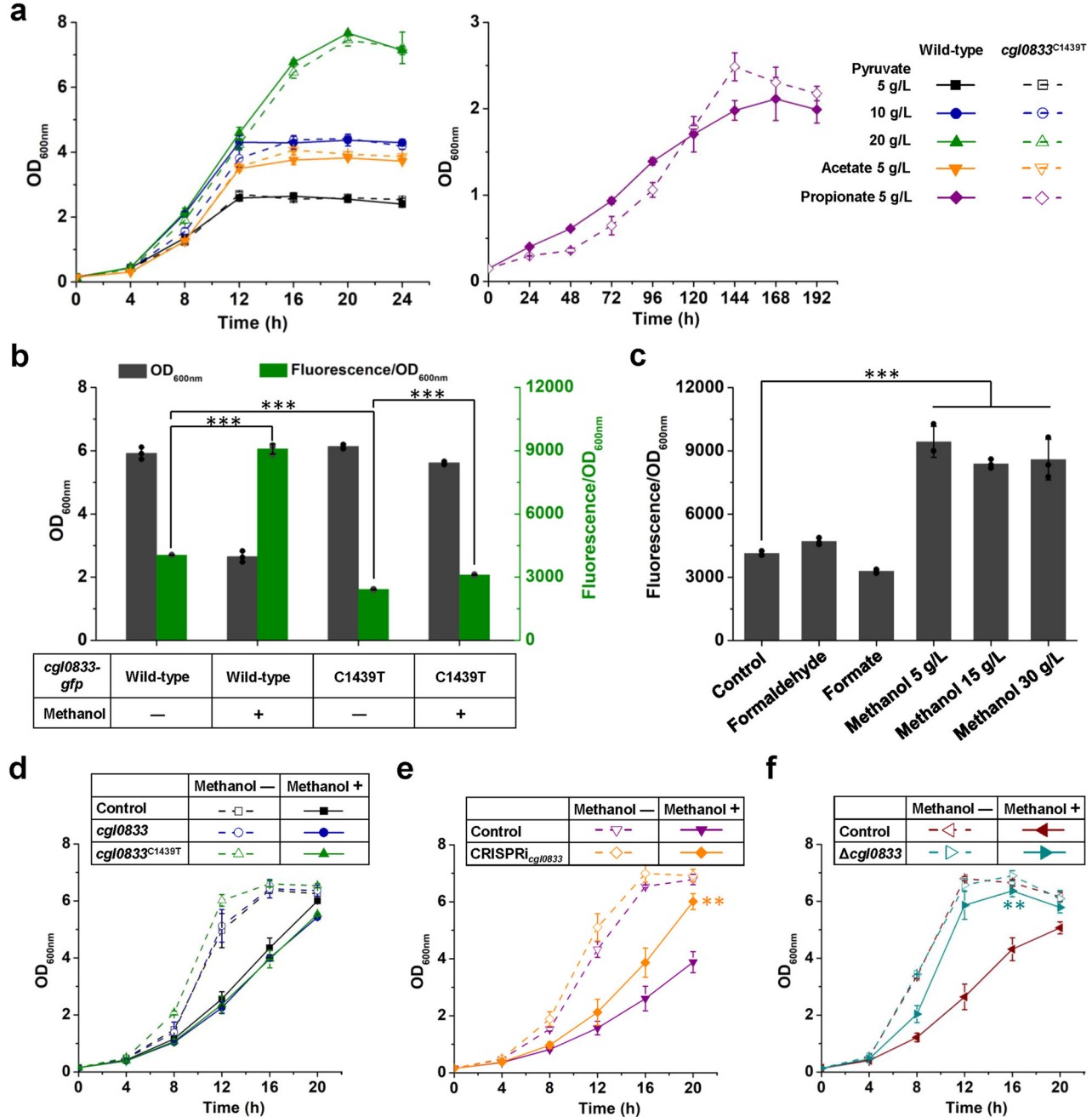

**Fig. 6 Effects of *cgl0833*$^{C1439T}$ mutation on its biological function, expression and methanol tolerance. a** Growth of *C. glutamicum* ATCC 13032 wild-type strain and *cgl0833*$^{C1439T}$ mutant strain on monocarboxylic acids. CGXII minimal media supplemented with different carbon sources were used for cultivation. **b** Effects of methanol addition and *cgl0833*$^{C1439T}$ mutation on *cgl0833* expression. *C. glutamicum* ATCC 13032 wild-type strain and *cgl0833*$^{C1439T}$ mutant strain with *gfp* fused to *cgl0833* were cultivated in CGXII minimal medium supplemented with 5 g/L glucose. Methanol (30 g/L) was added as required. ***$P < 0.001$, one-way ANOVA, $N = 3$; wild-type, methanol – vs. wild-type, methanol + , $P = 3.2 \times 10^{-6}$; wild-type, methanol –, vs. C1439T mutant, methanol –, $P = 5.7 \times 10^{-8}$; C1439T mutant, methanol – vs. C1439T mutant, methanol + , $P = 3.3 \times 10^{-6}$. **c** Induction of *cgl0833* by different C1 substrates. *C. glutamicum* ATCC 13032 wild-type strain with *gfp* fused to *cgl0833* was cultivated in CGXII minimal media supplemented with 5 g/L glucose as carbon source and different inducers. ***$P < 0.001$, one-way ANOVA, $N = 3$, $P = 2.5 \times 10^{-5}$. **d** Effects of *cgl0833* and *cgl0833*$^{C1439T}$ overexpression on methanol tolerance. **e** Effects of *cgl0833* knock-down on methanol tolerance. **$P < 0.01$, one-way ANOVA, $N = 3$; CRISPRi$_{cgl0833}$, methanol + vs. control, methanol + , $P = 0.0014$. **f** Effects of *cgl0833* knock-out on methanol tolerance. **$P < 0.01$, one-way ANOVA, $N = 3$; Δ*cgl0833*, methanol + vs. control, methanol + , $P = 0.0014$. CGXII minimal medium supplemented with 5 g/L glucose and 30 g/L methanol was used to cultivate *C. glutamicum* ATCC 13032 and derivatives. IPTG (0.1 mM) was added at 4 h to induce *cgl0833* or *cgl0833*$^{C1439T}$ overexpression and dCas9 expression. Values and error bars reflect the mean ± s.d. of three biological replicates ($N = 3$).

regeneration routes[23,24] have been mainly focused on by researchers to improve synthetic methylotrophy. In this study, we demonstrate that the underutilized tolerance engineering is a very useful strategy to enhance methanol bioconversion. Compared to the parent methylotrophic *C. glutamicum* strain[28], methanol-tolerant strains showed obviously increased growth (0.052 h$^{-1}$ vs. 0.030 h$^{-1}$), methanol consumption (203.75 mM vs. 96.90 mM), and methanol-based biosynthesis (230 mg/L L-glutamate vs. 90 mg/L) in the presence of high concentrations of methanol.

Methanol and its oxidized product formaldehyde at certain concentrations are toxic to both methylotrophic and non-methylotrophic microorganisms[29,30]. However, improved growth and methanol utilization were observed here when higher concentrations of methanol were provided as the carbon source. Transcriptome analysis revealed the possible mechanism behind this phenomenon, part of which is coincidentally consistent with previous studies aiming to rationally optimizing synthetic methylotrophy. First, glycolysis was repressed by increased methanol context. Woolston et al.[24] used iodoacetate, a potent inhibitor of glyceraldehyde-3-phosphate dehydrogenase, to block glycolysis, which drove more carbon flux into Ru5P regeneration and consequently accelerated formaldehyde assimilation. This suggests that an appropriate repression of glycolysis and the balance between RuMP and glycolysis pathways are prerequisites for fully synthetic methylotrophy. In addition to the substance metabolism, metabolism of energy and reducing power needs being rewired to suit methylotrophy. Among the generally up-regulated TCA cycle, the NAD-dependent Madh is exceptionally down-regulated. It has been shown that deactivation of Madh in methylotrophic *E. coli* benefited methanol metabolism[27]. Since methanol is more reduced than glucose and excess reducing power is generated by methanol metabolism, strategies for reducing NADH level are suggested to be explored for synthetic methylotrophy[21,24]. The attenuation of Madh activity and increase of NADH dehydrogenase activity observed here possibly rebalanced the intracellular NAD(H) level toward a high NAD$^+$/NADH ratio and facilitated efficient methanol oxidation.

Based on the genomic mutations accumulated in the methanol-tolerant strains, we can speculate the toxic mechanism of methanol to microorganisms. First, methanol acts as an analog of methanethiol and participates in the enzymatic reaction catalyzed by MetY which yields O-methyl-L-homoserine, an analog of L-methionine, and consequently produces dysfunctional proteins[48]. Disruption of Cgl0653 (MetY) activity in *C. glutamicum* via enzyme mutation or repressing gene expression could thus improve cellular tolerance to high concentrations of methanol. Second, as an organic solvent, methanol brings effects to cell wall and membrane. By analyzing the gene expression profiles of *Saccharomyces cerevisiae* treated with methanol, researchers discovered that genes induced by methanol stress mainly encode integral membrane proteins or proteins localized to the plasma membrane[29]. This consideration is further supported by the fact that membrane-bound protein Cgl2857 involved in synthesis of cell wall components mutated in a methanol-tolerance strain. Moreover, another membrane-bound protein Cgl0833 came into our sight because its mutation largely improved cellular tolerance to methanol but brought no obvious effect to its physiological function in monocarboxylic acid uptake[46]. Since expression of Cgl0833 is specifically induced by methanol, it may possess undiscovered functions that are related to resistance to high concentrations of methanol.

The evolved strain MX-14 still needs xylose as a co-substrate for methanol assimilation, which is a drawback compared to native methylotrophs that utilize methanol as the sole carbon source[50]. To reduce the cost of carbon source and improve economical performance, refined xylose is expected to be replaced with cheap raw sugar feedstocks such as lignocellulose hydrolysate, which contains various sugars including glucose and xylose[51]. Based on an in silico simulation and experimental verification, *rpiB*-deleted *C. glutamicum* strain cannot grow on glucose or xylose as the sole carbon source, whereas xylose or glucose can be utilized under a co-consumption regime with methanol (Supplementary Table 1 and Supplementary Fig. 6). Therefore, refined xylose can be potentially replaced with raw sugar feedstocks for methanol assimilation. Although strain MX-14 produced more L-glutamate than its parent strain MX-11 under high concentrations of methanol, the titer was still much lower than those of *C. glutamicum* using glucose[52] or native methylotroph *B. methanolicus* using methanol[53]. Future process engineering may further improve the L-glutamate production level of strain MX-14 from methanol and a cheap raw sugar feedstock.

In conclusion, tolerance engineering was proven to be an effective strategy for enhancing methanol bioconversion of synthetic methylotrophs. The uncovered metabolic response to high concentrations of methanol and genetic mutations conferring methanol resistance not only revealed the toxic mechanism of methanol but also provided direction for future engineering to improve methanol utilization.

## Methods

**Bacterial strains and growth conditions**. The bacterial strains used in this study are listed in Supplementary Table 2. *E. coli* strain DH5α was used for general cloning and cultivated at 37 °C and with shaking at 220 rpm in Luria–Bertani (LB) broth. Ampicillin (Amp, 100 μg/mL), kanamycin (Km, 50 μg/mL), or chloramphenicol (Cm, 20 μg/mL) was added as required. *C. glutamicum* ATCC 13032 and its derivatives were cultivated at 30 °C and with shaking at 220 rpm in LB medium or CGXII minimal medium[54] supplemented with glucose or organic acid as a carbon source. Methanol (5-30 g/L), formaldehyde (15 mg/L), or formate (5 g/L) was added prior to inoculation to provide a stress condition as required. The initial OD$_{600nm}$ was set as 0.1. Methanol-dependent *C. glutamicum* strains were cultivated at 30 °C and with shaking at 220 rpm in LB or CGXII minimal medium supplemented with methanol and xylose as co-substrates. Km (25 μg/mL) or Cm (5 μg/mL) was added for cultivating *C. glutamicum* strains as required. To avoid evaporation of methanol, the shake flasks were covered with a sealing membrane. The initial OD$_{600nm}$ was set as 0.5.

**Adaptive laboratory evolution**. ALE of *C. glutamicum* MX-11 was performed using CGXII minimal medium supplemented with 15 g/L methanol and 4 g/L xylose according to the procedure described previously[28]. Strain MX-11 was cultivated at 30 °C and with shaking at 220 rpm for ~96 h. Then, the culture was used as a seed to inoculate fresh medium with an initial OD$_{600nm}$ of 0.5, which was then incubated under the same conditions. At the certain passage that showed the best cell growth, the culture was diluted, plated on CGXII solid medium supplemented with 15 g/L methanol and 4 g/L xylose, and incubated at 30 °C. The clones that grew fast were cultivated in LB medium supplemented with 15 g/L methanol and 4 g/L xylose and stored for further experimental analysis.

**Quantitative measurement of methanol and xylose**. Methanol and xylose were measured according to the procedure described previously[28]. Cultures were harvested and centrifuged at 5000×*g* for 10 min and the supernatant was used for methanol and xylose measurement. For methanol analysis, the SBA-40 biosensor analyzer (Institute of Biology of Shandong Province Academy of Sciences, Shandong, China) equipped with an alcohol oxidase membrane was used. The analytical signal was given by quantifying the production of H$_2$O$_2$, which was generated by methanol oxidation catalyzed by the alcohol oxidase. Xylose was measured by using Prominence Ultra-Fast Liquid Chromatography (UFLC, Shimadzu, Japan) equipped with a refractive index detector and a Bio-Rad Aminex HPX-87H column (300 × 7.8 mm). A mobile phase of 5 mM H$_2$SO$_4$ at 55 °C was used at a flow rate of 0.5 mL/min. The injection volume was set at 10 μL. The substrate uptake rate (mM/h) was calculated using Eq. (1). The specific methanol uptake rate $q_M$ (mmol/gCDW·h) was calculated according to Eq. (2). $t$, $X_0$, and $\mu$ represent the time in hour, the initial biomass concentration, and the specific growth rate in h$^{-1}$, respectively. Cellular dry weight (CDW) was determined using a conversion factor of 0.30 gCDW/L·OD$_{600nm}$. The specific growth rate μ for strain MX-11 was obtained using exponential regression on growth data. The specific growth rate for each passage of ALE was calculated using the OD$_{600nm}$ values at the initial and final time points.

$$\frac{dS_i(t)}{dt} = \frac{s_i(t_2) - s_i(t_1)}{t_2 - t_1} \qquad (1)$$

$$q_{\mathrm{M}} = \frac{\mathrm{d}S_{\mathrm{M}}(t)/\mathrm{d}t}{X_0 \times e^{\mu t}} \qquad (2)$$

**L-Glutamate production and measurement**. *C. glutamicum* MX-14 was cultivated in modified CGXII minimal medium supplemented with 15 g/L methanol and 4 g/L xylose at 30 °C and with shaking at 220 rpm. Biotin was added to the medium to a final concentration of 0.5 μg/L. When OD$_{600nm}$ of the culture reached ~3.0, penicillin G was added to a final concentration of 60 U/mL to induce L-glutamate production. Samples were taken periodically and extracellular L-glutamate concentrations were quantified using an SBA-40D biosensor analyzer (Institute of Biology of Shandong Province Academy of Sciences, Shandong, China) equipped with a L-glutamate oxidase membrane according to the procedure described previously[28].

**Determination of ¹³C-labeled proteinogenic amino acids**. ¹³C-Methanol incorporation into proteinogenic amino acids was determined according to the procedure described previously[28]. *C. glutamicum* MX-14 was cultivated in CGXII minimal medium supplemented with 15 g/L ¹³C-methanol (99% atom enrichment, Sigma-Aldrich, USA) and 4 g/L non-labeled xylose. After 120 h of cultivation, cells were harvested by centrifugation at 5000×g for 10 min, washed twice with 50 mM potassium phosphate buffer (pH 7.4), resuspended in 6 M HCl, and transferred to glass screw-top GC vials. The vials were placed in a 105 °C oven for 24 h to hydrolyze biomass proteins into amino acids. The hydrolysates were centrifuged at 12,000×g for 10 min to remove solid particles in the hydrolysis solution and the supernatants were transferred into new centrifuge tubes for desiccation. For derivatization, 150 μL of 20 mg/mL methoxylamine hydrochloride in pyridine was added to samples, which were incubated at 30 °C for 90 min and vortexed occasionally. Then, 100 μL of *N*-methyl-*N*-trimethylsilyltrifluoroacetamide was added and the samples were incubated at 37 °C for another 60 min and vortexed occasionally. Next, the samples were centrifuged at 12,000×g for 10 min, and the supernatant was transferred to new GC vials. The derivatized amino acid samples were analyzed by GC/Q-TOF-MS using an Agilent 7890 A GC coupled with a 7200 Accurate-Mass Q-TOF (Agilent Technologies, Germany) and a DB-5MS Ultra Inert column (30 m × 0.25 mm, 0.25 μm film thickness, Agilent Technologies, USA). The injection volume was set at 1 μL. The oven temperature was programmed as follows: 60 °C for 1 min, 8 °C/min to 132 °C, 2 °C/min to 150 °C, 5 °C/min to 185 °C, 10 °C/min to 325 °C, 5 min hold. Mass spectra of amino acids were in the mass range of 50–650 *m/z* at an acquisition rate of 5 spectra/s. The temperatures of the ion source and transfer line were 250 °C and 290 °C, respectively. The electron ionization was carried out at 70 eV. Agilent Mass Hunter Qualitative Analysis Software was used for peak detection and mass spectral deconvolution. Annotation of amino acids was performed via matching their mass fragmentation patterns with those in the National Institute of Standards and Technology mass spectral library (match factor >80%).

**Transcriptome analysis**. *C. glutamicum* MX-14 was cultivated in CGXII minimal medium supplemented with 15 g/L methanol and 4 g/L xylose at 30 °C and with shaking at 220 rpm. Cells at the middle exponential phase (~72 h) were collected for RNA isolation. RNA preparation, library construction and sequencing on Illumina HiSeq were performed by Novogene (Tianjin, China). FASTQC software (v.0.10.1) was used to assess the quality of raw sequence reads. Based on the quality assessment results, low-quality reads and bases from both ends of raw Illumina reads were removed and trimmed using the NGSQC Toolkit (v.2.3.3) (-l 70, -s 25). After the high-quality reads were aligned against the *C. glutamicum* ATCC 13032 reference genome (GenBank accession number GCA_000011325.1) using BWA alignment software (v.0.7.17), the mapping results were sorted and indexed using SAM tools software (v.1.9). Raw read counts from BAM files were obtained using HTSeq (v.0.11.2) software. The raw-count table was further processed with the DESeq function of the DeSeq2 package (v.1.18.1) to obtain gene expression data. Genes with a false discovery rate (FDR) < 0.05 and log₂(Fold change)>0.5 or <−0.5 were considered to be differentially expressed. Pearson's linear correlation coefficients between variables were calculated using the R package 'stats' and plotted using 'corrplot'. Principal component analysis was performed using 'stats' package and plotted using 'ggord' package.

**Genome sequencing of evolved mutants**. Genomic DNAs of evolved *C. glutamicum* strains were extracted using Wizard Genomic DNA Purification Kit (Promega (Beijing) Biotech Co., China). Library construction and genome sequencing were performed by Berry Genomics (Beijing, China) by using Illumina Hiseq2500 sequencing platform. Quality assurance of the output was analyzed by using FastQC software (v.0.10.1) and NGSQC Toolkit software (v.2.3.3). BWA alignment software (v.0.7.17) and SAM tools software (v.1.9) were used for alignment and variant calling, respectively. Variations were annotated by using the SnpEff software (v.4.3i).

**Mutation and gene knock-in in *C. glutamicum***. The suicide plasmid pK18*mobsacB*[55] was used for integrating the single-site nucleotide mutation or *gfp* gene into

*C. glutamicum* via allele exchange. The plasmids used in this study and primers for plasmid construction are listed in Supplementary Table 2 and Supplementary Table 3, respectively. pK18-*cgl0653*$^{G1256A}$ containing a ~2-kb mutant fragment of *cgl0653* was constructed to integrate the *cgl0653*$^{G1256A}$ mutation into *C. glutamicum* chromosome. The *cgl0653* fragment containing G1256A mutation was amplified from the genomic DNA of *C. glutamicum* using the primer pair *cgl0653*$^{G1256A}$-F/*cgl0653*$^{G1256A}$-R and then ligated with the *Bam*HI linearized pK18mobsacB using the ClonExpress II One Step Cloning Kit (Vazyme Biotech, China). Primer synthesis and Sanger sequencing were performed by GENEWIZ (China). The resultant plasmid pK18-*cgl0653*$^{G1256A}$ was transferred into *C. glutamicum* ATCC 13032 and *C. glutamicum* MX-4 via electroporation for allelic exchange. Strain MX-4-*cgl0653*$^{G1256A}$ harboring the *cgl0653*$^{G1256A}$ mutation was transformed with pEC-XK99E-*mdh*$_{Bs2334}$-*hps*-*phi*$_{Bm}$ to generate strain MX-10-*cgl0653*$^{G1256A}$. Ribose and xylose were used as carbon sources during the manipulation process of strain MX-4. Integration of the rest single-site nucleotide mutations and *gfp* gene were conducted following a similar procedure.

**Gene overexpression in *C. glutamicum***. *E. coli-C. glutamicum* shuttle vector pEC-XK99E[56] were used for gene overexpression in *C. glutamicum*. Wild-type and mutant *cgl0653* and *cgl0833* genes were amplified from genomic DNAs of *C. glutamicum* strain ATCC 13032 and strain MX-14 using the primer pairs *cgl0653*-F/*cgl0653*-R and *cgl0833*-F/*cgl0833*-R, respectively. The PCR product was inserted into pEC-XK99E under the control of the isopropyl-β-D-thiogalactopyranoside (IPTG)-inducible promoter *P*$_{tac}$ using the ClonExpress II One Step Cloning Kit (Vazyme Biotech, China). The resultant plasmid was transformed into *C. glutamicum* ATCC 13032 by electroporation. IPTG (0.1 mM) was added at 4 h to induce gene overexpression.

**Gene knock-down in *C. glutamicum***. CRISPRi technique was used to knocking down genes in *C. glutamicum*. An all-in-one tool plasmid harboring dCas9 and gRNA expression cassettes was first constructed. dCas9 gene was amplified from a previously plasmid pdCas9[57] using the primer pair dCas9-F/dCas9-R. The PCR product was ligated with the *Hind*III and *Pst*I digested pnCas9(D10A)-AID-gRNA-*ccdB*$^{TS}$[58] to replace nCas9(D10A)-AID with dCas9 and produce pdCas9-gRNA-*ccdB*. Golden Gate assembly strategy was applied to construct CRISPRi tool plasmid harboring gRNA targeting a specific gene using pdCas9-gRNA-*ccdB* and a annealed double-stranded DNA from two 24-nt primers[57]. The CRISPRi tool plasmid was transformed into *C. glutamicum* ATCC 13032 by electroporation. IPTG (0.1 mM) was added at 4 h to induce dCas9 expression.

**Gene knock-out in *C. glutamicum***. pK18*mobsacB*[55] was used for gene knock-out in *C. glutamicum* via allele exchange. pK18-Δ*cgl0653* containing a mutant allele of *cgl0653* was constructed to knock out *cgl0653*. The mutant allele of *cgl0653* was generated by connecting a left and a right homologous flanks of *cgl0653*. First, the left and right flanks were amplified from the genomic DNA of *C. glutamicum* ATCC 13032 using the primer pairs Δ*cgl0653*-F1/Δ*cgl0653*-R1 and Δ*cgl0653*-F2/Δ*cgl0653*-R2, respectively (Supplementary Table 3). The two fragments were ligated with the *Bam*HI linearized pK18mobsacB to construct pK18-Δ*cgl0653*. pK18-Δ*cgl0833* was constructed following the same procedure and primer pairs Δ*cgl0833*-F1/Δ*cgl0833*-R1 and Δ*cgl0833*-F2/Δ*cgl0833*-R2. The resultant plasmid was transferred into *C. glutamicum* ATCC 13032 via electroporation for allelic exchange.

**Expression and purification of Cgl0653 and MetX**. *cgl0653* and *cgl0653*$^{G1256A}$ were amplified from genomic DNAs of *C. glutamicum* strain ATCC 13032 and strain MX-14, respectively, using the primer pair cgl0653-21a-F/cgl0653-21a-R (Supplementary Table 3). The PCR product was inserted between the *Nde*I and *Xho*I sites of pET-21a(+) and fused with a C-terminal His Tag using the ClonExpress II One Step Cloning Kit (Vazyme Biotech, China). Recombinant plasmids were transformed into *E. coli* BL21 (DE3) and heterogeneous expression and purification of protein were conducted according to the procedure described previously[22]. The resultant strains were cultivated in LB medium at 37 °C with shaking at 220 rpm. When the OD$_{600nm}$ reached 0.6–0.8, expression of heterologous genes was induced with 0.1 mM IPTG. After incubated at 16 °C for 14 h, cells were harvested and washed twice with 100 mM potassium phosphate buffer (pH 7.4). The cell pellet was resuspended in the same buffer and disrupted by sonication in an ice bath. The lysed cells were centrifuged at 10,000×g for 30 min at 4 °C. The enzyme was purified from the supernatant using a His-Trap column (GE Healthcare, USA) at 4 °C. MetX from *Leptospira meyeri*[59] fused with a N-terminal His Tag was expressed using pET-28(a) and purified with the same procedure as Cgl0653. Purified enzymes were analyzed by SDS-PAGE and protein concentration was determined with the BCA Protein Assay Kit (Thermo Fisher Scientific, USA).

**Enzyme activity assay of Cgl0653 and Cgl0653$^{G419D}$**. *O*-Methyl-L-homoserine formation activity of Cgl0653 and Cgl0653$^{G419D}$ was determined by a coupled enzymatic assay. The enzymatic reaction was performed in 400 μL volume of 100 mM potassium phosphate buffer (pH 7.4) containing 1 mM acetyl-CoA, 5 mM L-homoserine, 5 U MetX and 1 M methanol. The reaction was started by addition of 60 μg purified Cgl0653 or Cgl0653$^{G419D}$ enzyme and incubated at 30 °C. The reaction was stopped after 20 min and samples were withdrawn. *O*-Methyl-L-

homoserine was detected by LC-MS/MS according to the procedure described previously[28]. A Shimadzu Nexera Ultra-Performance Liquid Chromatography (UPLC) 30 A (Shimadzu, Japan) equipped with a SeQuant ZIC-HILIC column (100 mm × 2.1 mm, 3.5 µm, Merck, Germany) and an Applied Biosystem Triple-TOF$^{TM}$ 5600 mass spectrometer with a resolution of 30,000 FWHM (Applied Biosystem, USA) and negative electrospray ionization (ESI) mode was used for LC-MS/MS analysis. The mobile phases included A phase (10 mmol/L $(NH_4)_2COOH$) and B phase (100% acetonitrile) and the flow rate was set at 0.2 mL/min. The LC gradient was 0–3 min, 90% B; 3–6 min, 90%–60% B; 6–25 min, 60%–50% B; 25–30 min, 50% B; 30–30.5 min, 50%–90% B; 30.5–38 min, 90% B. The flow rate was set at 0.2 mL/min. The mass spectra were obtained from ESI negative mode of −35 eV with a scan range of 30–1200 $m/z$. Moreover, mass accuracy was calibrated by automated calibrant delivery system (AB Sciex, Canada) interfaced to the second inlet of the DuoSpray source. The injection volume was set at 5 µL.

**Homologous modeling and flexible docking**. The model structure of the wild-type Cgl0653 was constructed with the crystal structure of O-acetyl-L-homoserine sulfhydrylase from *Mycobacterium marinum* ATCC BAA-535 (PDB ID: 4KAM) as a template (54% sequence identity with Cgl0653) using Discovery Studio 4.1 software (Dassault Systèmes, BIOVIA Corp., USA). Flexible dockings of pyridoxal-5-phosphate (PLP), methanol, and O-acetyl-L-homoserine into the active site were performed with Discovery Studio 4.1 software. Other parameters were kept as default settings in flexible docking.

**Peptide uptake and proline export assay**. Pregrown *C. glutamicum* ATCC 13032 wild-type cells and *cgl0833*$^{C1439T}$ mutant cells with LBG medium supplemented with 5 g/L methanol were harvested and washed twice with ice-cold CGXII medium supplemented with 5 g/L glucose. The peptide uptake and amino acid excretion were then initiated by resuspending the cells in prewarmed CGXII medium (30 °C) containing 5 g/L glucose and 2 mM Phe-Pro peptide. The resultant cell density (OD$_{600nm}$) was 10.0. The cells were incubated at 30 °C and with shaking at 220 rpm. Samples were taken every 15 min and extracellular amino acids were quantified using Prominence UFLC (Shimadzu, Japan) equipped with a Zorbax Eclipse AAA column (4.6 mm × 150 mm, 5 µm) and a UV detector[60]. A gradient of 50 mM sodium acetate buffer at pH 6.4 with a gradient solution containing acetonitrile-water (50%, vol/vol) was used as the eluent. Amino acids were detected as their 2,4-dinitrofluorobenzene derivatives at 360 nm by following the precolumn derivation method.

**Assay of *cgl0833* expression by detecting GFP fluorescence**. *C. glutamicum* strains with *gfp* fused to *cgl0833* or *cgl0833*$^{C1439T}$ were cultivated in CGXII minimal media supplemented with 5 g/L glucose as a carbon source. Different inducers, methanol (5–30 g/L), formaldehyde (15 mg/L), or formate (5 g/L), were added prior to inoculation. The cultures were incubated at 30 °C and with shaking at 220 rpm for 12 h. Cells were then harvested by centrifugation at 5000×g for 10 min, washed once, and re-suspended in 100 mM potassium phosphate buffer (pH 7.4). GFP fluorescence intensity was determined using a SpectraMax M5 microplate reader (Molecular Devices, USA, λ excitation = 488 nm, λ emission = 520 nm).

**Measurement of *cgl0833* transcription by qPCR**. *C. glutamicum* ATCC 13032 was cultivated in CGXII minimal media supplemented with 5 g/L glucose as a carbon source. Different inducers, methanol (5 g/L), formaldehyde (15 mg/L), or formate (5 g/L), were added prior to inoculation. The cultures were incubated at 30 °C and with shaking at 220 rpm. Cells were collected in the mid-exponential phase (OD$_{600nm}$ ≈ 3) and total RNAs were isolated using the RNAprep Pure Cell/Bacteria Kit (Tiangen Biotech, China). After treated with DNase I (Tiangen Biotech, China), total RNA samples were used to synthesize cDNAs using random primers and Fast Quant RT Kit (Tiangen Biotech, China). The resultant cDNAs were used as templates for qPCR analysis. The total RNA samples were also used as templates for qPCR to confirm that genomic DNA contamination during total RNA extraction was minimal. Specific primers for qPCR were designed using Beacon Designer software v7.7 (PREMIER Biosoft International, USA) (Supplementary Table 3). qPCR was performed using the SuperReal Premix SYBR Green Kit (Tiangen Biotech, China) and Applied Biosystems® 7500 Real-Time PCR System (Thermo Fisher Scientific, USA) according to the manufacturer's instructions.

**In silico analysis of *rpiB* deletion on cell growth**. The genome-scale metabolic model *i*CW773[61] was used to predict the growth of *C. glutamicum* by performing flux balance analysis (FBA)[62]. Since the *i*CW773 model does not contain any methanol assimilation pathways, the reactions catalyzed by Mdh (methanol + NAD$^+$ <=> formaldehyde + NADH + H$^+$), Hps (ribulose-5-phosphate + formaldehyde <=> hexulose-6-phosphate), and Phi (hexulose-6-phosphate <=> fructose-6-phosphate) were added to the *i*CW773 model. To simulate *rpiB* deletion, the reaction catalyzed by RpiB (ribose-5-phosphate <=> ribulose-5-phosphate) was turned off in the *i*CW773 model. Simulations were performed using the COBRApy toolbox[63]. Uptake rate of each carbon source was set as 1 mmol/gCDW·h.

**Statistics and reproducibility**. Three biological replicates were conducted for all the experiments. Data are showed as mean ± s.d. (*N* = 3). A one-way ANOVA was used to assess significance between more than two groups (*N* = 3).

**Reporting summary**. Further information on research design is available in the Nature Research Reporting Summary linked to this article.

## Data availability
The sequencing data were deposited into the Sequence Read Archive (SRA) database (accession number: PRJNA615290) at the National Center for Biotechnology Information (NCBI). All the other data and materials that support the findings of this study are available from the corresponding author on reasonable request.

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

## Acknowledgements
This work was supported by National Key R&D Program of China (2018YFA0901500 and 2018YFA0903600), National Natural Science Foundation of China (31700044 and 21908239), and Key R&D Plan of Shandong Province (2017CXGC1103). We thank Mr. Yonghong Yao at Technology Support Center of Tianjin Institute of Industrial Bio-technology for help with GC/Q-TOF-MS analysis.

## Author contributions
Y.W., P.Z., J.S., and Y.M. conceived and initiated the project. Y.W. designed the experiments. Y.W., L.F., P.T., J.L., K.Z., N.G., and Z.Z. carried out the experiments. X.N., J.F., Q.Y., and H.M. performed bioinformatic analysis. Y.W., L.F., and P.T. analyzed the data. Y.W. wrote the initial manuscript draft and all authors contributed to discussion and writing of the final manuscript.

## Competing interests
The authors declare no competing interests.
