## [Peer Review File · Communications Biology]

Reviewers' comments:

Reviewer #1 (Remarks to the Author):

Wang and co-workers performed adaptive laboratory evolution of *C. glutamicum* towards increased robustness in the presence of high methanol concentrations and obtained variants exhibiting higher robustness and higher methanol consumption capabilities. The authors performed transcriptome analyses and genome sequencing to discover underlying changes in gene transcription and/or enzyme activities.

The manuscript appears to be well organized, but contains many typos and grammatical errors.

In the abstract the authors promise to "provide a new strategy to enhance methanol bioconversion of synthetic methylotrophs by improving cellular tolerance to methanol". Well the presented strategy has been presented before – by the same authors. Essentially, this manuscript is a follow-up study to "Tuyishime, P. et al. Engineering *Corynebacterium glutamicum* for methanol-dependent growth and glutamate production. *Metab. Eng.* 49, 220–231 (2018)." by the same authors in which the same techniques were applied.

The growth rate of *C. glutamicum* on methanol/xylose could only be increased from 0.03 h⁻¹ to 0.052 h⁻¹ and the final biomass is still very low. This is also the reason why for the most part, improvements throughout the manuscript are given as "fold improvement". However, important parameters are still way too low to really think about any application of this strain. In general, the experiments appear to be well executed. However, sometimes experiments were performed for no obvious reasons, e.g. why would somebody calculate a homology model of an enzyme before in vitro enzyme assays were conducted?

The materials and methods section well organized but important pieces of information are often missing, which will make it difficult for researchers to redo the experiment. E.g. how much Cgl0653 or Cgl0653-G419D was added to start the enzyme assays?

Taken together, I do not think that the manuscript is of great relevance for the readership of this journal. I suggest to send it to a more specialized journal.

Reviewer #2 (Remarks to the Author):

The manuscript "Improving methanol tolerance enhances methanol conversion in engineered *Corynebacterium glutamicum*" described about improvement of methanol utilization ability of previously developed engineered methylotrophic *C. glutamicum*. Transcriptome and genome analysis were performed to clarify the mechanism for improving methanol utilization. The paper showed some results, but there are some criticisms.

(1) The engineered *C. glutamicum* required xylose as co-substrate for the production of glutamate. However, a technology platform for the production of useful chemicals by engineered microbes using methanol as sole carbon source has been reported (doi.org/10.1007/s11274-019-2610-4). Can the present engineered *C. glutamicum* produce

useful chemicals from methanol as the sole carbon source? This is because, to prepare pure xylose from renewable resource will be difficult. It will contain various sugars such as glucose, which is preferably assimilated by *C. glutamicum*.

(2) The production of glutamate has been realized by bacteria such as *Bacillus methanolicus* and *Pseudomonas insueta* (doi.org/10.1007/s00253-006-0757-z). How does the glutamate productivity of present *C. glutamicum* compare to previous reports? The author should discuss about that.

(3) Page 23, Line 404: The author evaluated the fluorescence intensity to assess the expression level of *cgl0833*-GFP fusion protein. However, the analysis reflects the activity, not the expression of *cgl0833*. Real-time PCR analysis is suitable for assessing gene expression. Why did the authors evaluate the fluorescence of the GFP fusion protein for expression analysis?

(4) Fig. 2: Why do MX-14 mutants grow well at high methanol concentrations? In other words, why does MX-14 stop growing at around OD600 of 4.3 with 4g/L of methanol? At that time, methanol was not exhausted. Besides, methanol was not completely consumed in any methanol concentration medium. Why is this phenomenon observed? This is an obvious drawback and needs to be discussed.

Reviewer #3 (Remarks to the Author):

The authors reported evolution and improvement of an engineered *Corynebacterium glutamicum* methanol auxotroph strain. They tried to improve methanol tolerance from 4g/L to 15g/L but also found out that growth was improved, as well as methanol uptake. They also used transcriptomics and NGS to identify the mutations and expression profile change. They specified two gene mutations, *cgl0653* and *cgl0883*. *cgl0653* utilizes methanol as a substrate to produce faulty amino acid which cause global protein structure disruption. The function of *cgl0683* is unknown, but they found out that lower expression of *cgl0883* benefits methanol growth.

Below are some comments

1. The authors should acknowledge that methanol tolerance in *E. coli* is higher than WT *Corynebacterium glutamicum*. Also authors should at discuss what is the pros of utilizing *Corynebacterium* as the heterologous host, compared to others (E.g *E. coli*)
2. In line 143, theoretically, the *rpiB* auxotroph should uptake methanol plus xylose in a 1:1 molar ratio. What is the explanation for the increased ratio of methanol utilization? This result seems inconsistent with the pathway proposed. Either the data or the pathway is questionable.
3. As mentioned in 2, The authors should implement a C13 labeling experiment to indicate biomass ratio coming from methanol. It may also strengthen their claim that methanol

uptake indeed increased. Mentioned in 2, possible ways of methanol incorporation may be caused by other factors. If so, the authors should also perform C13 experiments to prove that it comes from methanol.

4. The argument in Line 414 about formate possibly as a potential inducer is also questionable, as the authors already knocked out the formaldehyde detoxifying pathway.

5. The CRISPR data was well presented in the paper. It may be interesting if the authors try to directly knockout the genes that were in question. Specifically, as they stated lower expression of Cgl0833 may be beneficial, it will be interesting if they can characterize the result of the gene being entirely knocked out.

Responses to the reviewers' comments point-by-point

Thank you for the reviewers' comments, which are very helpful for us to improve our manuscript. We carefully considered all the comments and revised the manuscript accordingly. A marked-up manuscript with the changes highlighted in red have been uploaded. We hope that the answers and the changes in the manuscript will enable our manuscript to win your satisfaction.

Reviewer #1

Wang and co-workers performed adaptive laboratory evolution of C. glutamicum towards increased robustness in the presence of high methanol concentrations and obtained variants exhibiting higher robustness and higher methanol consumption capabilities. The authors performed transcriptome analyses and genome sequencing to discover underlying changes in gene transcription and/or enzyme activities.

The manuscript appears to be well organized, but contains many typos and grammatical errors.

Response:

Thanks for your helpful comments. We are sorry for the typos and grammatical errors in our manuscript. We have carefully gone through the manuscript to correct them.

In the abstract the authors promise to “provide a new strategy to enhance methanol bioconversion of synthetic methylotrophs by improving cellular tolerance to methanol”. Well the presented strategy has been presented before – by the same authors. Essentially, this manuscript is a follow-up study to “Tuyishime, P. et al. Engineering Corynebacterium glutamicum for methanol-dependent growth and glutamate production. Metab. Eng. 49, 220–231 (2018).” by the same authors in which the same techniques were applied.

Response:

Thanks for your helpful comments. In the present and our previous studies, adaptive laboratory evolution was both used. However, the core strategy used in this study is tolerance engineering, which has not been used for enhancing methanol bioconversion in either our previous study or other studies on synthetic methylotrophy. Previous efforts are mostly made in engineering metabolic pathways or enzymes. Tolerance engineering that has long been neglected was proven to an effective strategy for

enhancing methanol bioconversion in this study. To perform tolerance engineering, we used one of our previously engineered methylotrophic *C. glutamicum* strains (Tuyishime, P. et al. *Metab. Eng.* 49, 220–231 (2018)) as the starting strain due to its capability to utilize methanol as a carbon source.

The growth rate of C. glutamicum on methanol/xylose could only be increased from 0.03 h⁻¹ to 0.052 h⁻¹ and the final biomass is still very low. This is also the reason why for the most part, improvements throughout the manuscript are given as “fold improvement”. However, important parameters are still way too low to really think about any application of this strain.

Response:

Thanks for your helpful comments. We agree with you that now the strain cannot meet the demand of industrial applications. However, this strain co-utilizes methanol and xylose at a ratio of 7:1 in minimal medium, which already outperforms most synthetic methylotrophs based on *E. coli* or *C. glutamicum* (as reviewed in *Trends. Biotechnol.* DOI:10.1016/j.tibtech.2019.12.013 (2020) and *Biotechnol. Adv.* DOI: 10.1016/j.biotechadv.2019.107467 (2019)). As you mentioned, we still have a long way to go before synthetic methylotrophs can be used for industrial bioconversion of methanol into useful chemicals, but we think every step matters.

In general, the experiments appear to be well executed. However, sometimes experiments were performed for no obvious reasons, e.g. why would somebody calculate a homology model of an enzyme before in vitro enzyme assays were conducted?

Response:

Thanks for your helpful comments. We realize that the logic of homology modeling and enzyme activity assay is confusing. Actually, we did these two experiments at the same time. To make presentation of results more logical, we switched the order of homology modeling and enzyme activity assay in the revised manuscript. We have gone through the manuscript to avoid such confusion.

Please refer to P14, L293: “To better understand why the evolved mutation decreased activity, a homology model of *C. glutamicum* Cgl0653 was constructed with a close homologue. However, the G419 is predicted to localize to the region far from the binding pocket of co-factor pyridoxal-5-phosphate and substrates methanol

and *O*-acetyl-L-homoserine, which unlikely directly affects binding of co-factor or substrates (Supplementary Fig. 2).”

The materials and methods section well organized but important pieces of information are often missing, which will make it difficult for researchers to redo the experiment. E.g. how much Cgl0653 or Cgl0653-G419D was added to start the enzyme assays?

Response:

Thanks for your helpful comments. For each assay, 60 µg Cgl0653 or Cgl0653^{G419D} was added. We have added more details in the revised “Methods” section and avoided ‘as previously described’.

Taken together, I do not think that the manuscript is of great relevance for the readership of this journal. I suggest to send it to a more specialized journal.

Response:

Thanks for your comments. We have revised our manuscript according to your helpful comments and hope the revision can win your satisfaction. According to the policy of the journal *Communications Biology*, research papers published by the journal represent significant advances bringing new biological insight to a **specialized area of research**. We think our manuscript falls into the scope of the journal and will receive attention from researches working on synthetic biology, metabolic engineering, and bioconversion of single carbon (C1) feedstocks.

Reviewer #2:

The manuscript “Improving methanol tolerance enhances methanol conversion in engineered Corynebacterium glutamicum” described about improvement of methanol utilization ability of previously developed engineered methylotrophic C. glutamicum. Transcriptome and genome analysis were performed to clarify the mechanism for improving methanol utilization. The paper showed some results, but there are some criticisms.

(1) The engineered C. glutamicum required xylose as co-substrate for the production of glutamate. However, a technology platform for the production of useful chemicals by engineered microbes using methanol as sole carbon source has been reported

(doi.org/10.1007/s11274-019-2610-4). Can the present engineered *C. glutamicum* produce useful chemicals from methanol as the sole carbon source? This is because, to prepare pure xylose from renewable resource will be difficult. It will contain various sugars such as glucose, which is preferably assimilated by *C. glutamicum*.

Response:

Thanks for your helpful comments. Yes, native methylotrophs that can utilize methanol as the sole carbon source can be engineered to produce useful chemicals from methanol. However, considering the advantages of industrial platform microorganisms like *E. coli* and *C. glutamicum* in terms of advanced genetic engineering tools, wide product spectrum, etc., synthetic methylotrophs are still of great interest. The present synthetic methylotrophs, including the engineered *C. glutamicum* strain reported in this study, still cannot utilize methanol as the sole carbon source (as reviewed in Trends. Biotechnol. DOI:10.1016/j.tibtech.2019.12.013 (2020) and Biotechnol. Adv. DOI: 10.1016/j.biotechadv.2019.107467 (2019)). Based on an *in silico* simulation and experimental verification, *rpiB*-deleted *C. glutamicum* strain cannot grow on xylose or glucose as the sole carbon source. However, xylose or glucose can be utilized by *rpiB*-deleted strain under a co-consumption regime with methanol (data were added in the Supplementary Information as Supplementary Table 2 and Supplementary Fig. 5). Therefore, purified xylose can be potentially replaced with raw sugar feedstocks containing various sugars like glucose and xylose, which will be co-utilized with methanol. The reference you mentioned has also been cited in the revised manuscript to discuss the limitation and perspective of this study.

Please refer to P19, L413: “The evolved strain MX-14 still needs xylose as a co-substrate for methanol assimilation, which is a drawback compared to native methylotrophs that utilize methanol as the sole carbon source⁴⁹. To reduce the cost of carbon source and improve economical performance, refined xylose is expected to be replaced with cheap raw sugar feedstocks such as lignocellulose hydrolysate, which contains various sugars including glucose and xylose⁵⁰. Based on an *in silico* simulation and experimental verification, *rpiB*-deleted *C. glutamicum* strain cannot grow on glucose or xylose as the sole carbon source, whereas xylose or glucose can be utilized under a co-consumption regime with methanol (Supplementary Table 2 and Supplementary Fig. 5). Therefore, refined xylose can be potentially replaced with raw sugar feedstocks for methanol assimilation.”

Please refer to P31, L682: “*In silico* analysis of *rpiB* deletion on cell growth. The

genome-scale metabolic model *iCW77361* was used to predict the growth of *C. glutamicum* by performing flux balance analysis (FBA)⁶². Since the *iCW773* model does not contain any methanol assimilation pathways, the reactions catalyzed by Mdh (methanol + NAD⁺ \rightleftharpoons formaldehyde + NADH + H⁺), Hps (ribulose-5-phosphate + formaldehyde \rightleftharpoons hexulose-6-phosphate), and Phi (hexulose-6-phosphate \rightleftharpoons fructose-6-phosphate) were added to the *iCW773* model. To simulate *rpiB* deletion, the reaction catalyzed by RpiB (ribose-5-phosphate \rightleftharpoons ribulose-5-phosphate) was turned off in the *iCW773* model. Simulations were performed using the COBRApy toolbox⁶³. Uptake rate of each carbon source was set as 1 mmol/gCDW·h.”

Please refer to Supplementary Table 2:

Supplementary Table 2 *In silico* analysis of *rpiB* deletion on cell growth with different carbon sources^a

Carbon source ^b	Cell growth (h ⁻¹)
Xylose	0
Glucose	0
Methanol and xylose	0.081
Methanol and glucose	0.116

^aThe reaction catalyzed by RpiB (R5P \rightleftharpoons Ru5P) was deleted from the genome-scale metabolic model of *C. glutamicum* ATCC13032, *iCW773*¹.

^bUptake rate of each carbon source was set as 1 mmol/gCDW·h.

Please refer to Supplementary Fig. 5:

Supplementary Fig. 5 Effects of *rpiB* deletion on cell growth on xylose and glucose. *C. glutamicum* strains were cultivated using CGXII minimal medium supplemented with 4 g/L glucose or 4 g/L xylose as the carbon source. Error bars indicate standard deviations from three parallel experiments.

(2) *The production of glutamate has been realized by bacteria such as Bacillus methanolicus and Pseudomonas insueta (doi.org/10.1007/s00253-006-0757-z). How does the glutamate productivity of present C. glutamicum compare to previous reports? The author should discuss about that.*

Response:

Thanks for your helpful comments. Native methylotroph *Bacillus methanolicus* can produce over 50 g/L glutamate in fed-batch fermentation. Because the engineered *C. glutamicum* was cultivated in shake flasks and with limited carbon sources, the glutamate production level was lower than 1 g/L. We used glutamate as a target product to demonstrate methanol bioconversion could be enhanced by improving cellular tolerance to methanol. We have cited the mentioned reference and discussed the glutamate production level in the revised manuscript.

Please refer to P19, L422: “Although strain MX-14 produced more L-glutamate than its parent strain MX-11 under high concentrations of methanol, the titer was still much lower than those of *C. glutamicum* using glucose⁵¹ or native methylotroph *B. methanolicus* using methanol⁵². Future process engineering may further improve the L-glutamate production level of strain MX-14 from methanol and a cheap raw sugar feedstock.”

(3) *Page 23, Line 404: The author evaluated the fluorescence intensity to assess the expression level of cgl0833-GFP fusion protein. However, the analysis reflects the activity, not the expression of cgl0833. Real-time PCR analysis is suitable for assessing gene expression. Why did the authors evaluate the fluorescence of the GFP fusion protein for expression analysis?*

Response:

Thanks for your helpful comments. Since only one mutation C1439T was detected in *cgl0833* gene and no mutation was found in the promoter region of *cgl0833*, we speculated that this mutation would not affect the transcription of *cgl0833*. However, nucleotide changes may influence the translation of mRNA and consequently influence the expression level (Science 324, 255–258 (2009)). Therefore, we fused a *gfp* gene to *cgl0833* and evaluate the fluorescence of the GFP fusion protein for expression analysis.

We agree with you that real-time PCR analysis is suitable for assessing regulation of

gene transcription. Therefore, real-time PCR analysis was performed to investigate the induction of *cgl0833* upon treatments of methanol, formaldehyde, and formate. The results were added in the revised manuscript as Supplementary Fig 4. Consistent with the fluorescence assay of Cgl0833-GFP fusion, addition of methanol increased the transcription level of *cgl0833* by 4.70-fold, while no induction on *cgl0833* transcription by formaldehyde and formate was observed.

Please refer to P16, L347: “To test whether *cgl0833* was induced by methanol at transcription level, quantitative PCR (qPCR) was carried out. Indeed, methanol addition increased mRNA level of *cgl0833* by 4.70-fold, whereas formaldehyde or formate did not cause induction effects (Supplementary Fig. 4), which was consistent with the GFP fluorescence assay.”

Please refer to Supplementary Fig. 4:

Supplementary Fig. 4 Relative transcription level of *cgl0833* in *C. glutamicum* ATCC 13032 under treatment with methanol (5 g/L), formaldehyde (15 mg/L), or formate (5 g/L). Error bars indicate standard deviations from three parallel experiments. *P* value was calculated using *t*-test (**P*<0.05).

(4) Fig. 2: Why do MX-14 mutants grow well at high methanol concentrations? In other words, why does MX-14 stop growing at around OD600 of 4.3 with 4g/L of methanol? At that time, methanol was not exhausted. Besides, methanol was not completely consumed in any methanol concentration medium. Why is this phenomenon observed? This is an obvious drawback and needs to be discussed.

Response:

Thanks for your helpful comments. Transcriptome analysis revealed the possible mechanism of better cell growth of strain MX-14 at high methanol concentrations, part of which is coincidentally consistent with previous studies aiming to rationally optimizing synthetic methylotrophy. We have discussed the possible mechanism in the second paragraph of discussion section.

Strain MX-14 stopped growing at approximately 120 h, when xylose was exhausted but methanol was not. Because of deletion of *rpiB*, *C. glutamicum* cannot grow using methanol as the sole carbon source. Once xylose is exhausted, methanol consumption and cell growth ceased soon. We have added discussion on this phenomenon in the revised manuscript.

Please refer to P7, L132: “It was noticed that cell growth ceased at approximately 120 h when xylose was exhausted but methanol was not (Fig. 2). Deactivation of ribose phosphate isomerase (RpiB) coupled cell growth with methanol and xylose co-utilization. However, when xylose was exhausted, cells cannot maintain growth with methanol as the sole carbon source.”

Reviewer #3:

The authors reported evolution and improvement of an engineered Corynebacterium glutamicum methanol auxotroph strain. They tried to improve methanol tolerance from 4g/L to 15g/L but also found out that growth was improved, as well as methanol uptake. They also used transcriptomics and NGS to identify the mutations and expression profile change. They specified two gene mutations, cgl0653 and cgl0883. cgl0653 utilizes methanol as a substrate to produce faulty amino acid which cause global protein structure disruption. The function of cgl0833 is unknown, but they found out that lower expression of cgl0883 benefits methanol growth.

Below are some comments

1. The authors should acknowledge that methanol tolerance in E. coli is higher than WT Corynebacterium glutamicum. Also authors should at discuss what is the pros of utilizing Corynebacterium as the heterologous host, compared to others (E.g E coli)

Response:

Thanks for your helpful comments. In most studies on synthetic methylotrophy, 250 mM (8 g/L) or lower concentrations of methanol was used for growth test of both

methylotrophic *E. coli* (Ref. 13,15,18,23,25,31) and *C. glutamicum* strains (Ref. 10,27,32), except that a methanol-essential *E. coli* strain reported by Meyer and colleagues could grow with 500 mM methanol (Ref. 26). Since there is no detailed research on the methanol tolerance in *E. coli*, we are not sure whether *E. coli* has higher tolerance to methanol than *C. glutamicum*. We have modified the introduction section and cited all the relevant references to objectively describe the methanol concentration used for synthetic methylotrophs. The pros of utilizing *C. glutamicum* as the heterologous host was also introduced in the revised manuscript.

Please refer to P4, L79: “Therefore, methanol is usually used at a concentration lower than 250 mM (8 g/L) for representative native methylotrophs *B. methanolicus* MGA3²⁹ and *Methylobacterium extorquens* AM1³⁰ and methylotrophic *E. coli*^{13,15,18,23,25,31} and *C. glutamicum* strains^{10,27,32}, except that a methanol-essential *E. coli* strain could tolerate 500 mM methanol²⁶.”

Please refer to P5, L84: “*C. glutamicum* is one of the most important industrial workhorses due to its GRAS status (generally regarded as safe), relatively few growth requirements, and ability to produce and secrete large amounts of amino acids³³.”

2. In line 143, theoretically, the *rpiB* auxotroph should uptake methanol plus xylose in a 1:1 molar ratio. What is the explanation for the increased ratio of methanol utilization? This result seems inconsistent with the pathway proposed. Either the data or the pathway is questionable.

Response:

Thanks for your helpful comments. If all the formaldehyde acceptor Ru5P is totally produced from xylose metabolism, the *rpiB* auxotroph should uptake methanol plus xylose in a 1:1 molar ratio. However, Ru5P can also be regenerated from the engineered RuMP cycle, which incorporates carbons from methanol into Ru5P. We believe this is the reason why methanol and xylose were utilized at a mole ratio higher than 1:1. Both our previous results (Metab. Eng. 49, 220–231 (2018)) and those obtained by Meyer and colleagues (Nat. Commun. 9, 1508 (2018)) have demonstrated co-utilization of methanol with a co-substrate (xylose or gluconate) at a molar ratio higher than 1:1.

We also performed a ¹³C-labelling experiment using ¹³C-methanol and non-labeled xylose to explain the phenomenon. Completely ¹³C-labeled L-glycine, L-alanine, L-serine, L-threonine, L-aspartate, L-glutamate, L-proline, and L-valine were detected

in biomass. Production of these multiple-carbon labeled amino acids suggest that the formaldehyde acceptor is partially provided through the cycling RuMP pathway, not only from exogenous xylose. The results were consistent with the exceeded equimolar consumption of methanol and xylose mentioned above. We have added these results in the revised manuscript as Figs. 2e and 2f.

Please refer to P7, L146: “¹³C-methanol labeling approach has been applied to measure methanol incorporation into cellular biomass³⁴. To further demonstrate the enhanced methanol assimilation, biomass samples of strain MX-14 cultivated with 15 g/L ¹³C-methanol and 4 g/L non-labeled xylose were collected, hydrolyzed and analyzed for ¹³C-labeling in proteinogenic amino acids using GC/Q-TOF-MS. All the detected amino acids were ¹³C-labeled, including completely labeled L-glycine, L-alanine, L-serine, L-threonine, L-aspartate, L-glutamate, L-proline, and L-valine (Fig. 2e). Production of these multiple-carbon labeled amino acids suggest that the formaldehyde acceptor is partially provided through the cycling RuMP pathway, not only from exogenous xylose. The results were consistent with the exceeded equimolar consumption of methanol and xylose mentioned above. The average carbon labeling levels of these amino acids were between 20% and 30% (Fig. 2f), which were 1.20- to 1.70-fold higher than those of the parent strain MX-11 with 4 g/L methanol²⁷, indicating more methanol was assimilated into biomass. Taken together, the results demonstrate that improving methanol tolerance is an effective strategy to enhance methanol bioconversion.”

Please refer to Figs. 2e and 2f:

Fig. 2 e Relative abundance of proteinogenic amino acid mass isotopomers. f Average ¹³C-labeling of proteinogenic amino acids. *C. glutamicum* MX-14 was cultivated in CGXII minimal medium supplemented with 15 g/L ¹³C-methanol and 4 g/L

non-labeled xylose. Cells were collected at 120 h for ¹³C-labeling analysis. Error bars indicate standard deviations from three parallel experiments.

3. As mentioned in 2, The authors should implement a C13 labeling experiment to indicate biomass ratio coming from methanol. It may also strengthen their claim that methanol uptake indeed increased. Mentioned in 2, possible ways of methanol incorporation may be caused by other factors. If so, the authors should also perform C13 experiments to prove that it comes from methanol.

Response:

Thanks for your helpful comments. We have performed a ¹³C-labing experiment using ¹³C-methanol and non-labeled xylose as required. The results support the increased methanol uptake. Please refer to our response to your comment 2.

4. The argument in Line 414 about formate possibly as a potential inducer is also questionable, as the authors already knocked out the formaldehyde detoxifying pathway.

Response:

Thanks for your helpful comments. Because methanol is an indispensable carbon source for the growth of strain MX-14, we cannot analyze the induction of *cgl0833* by adding or removing methanol from the medium. Therefore, induction of *cgl0833* was analyzed in wild-type *C. glutamicum* ATCC 13032 strain, the formaldehyde detoxification pathway of which is intact. Methanol can be converted to formaldehyde and further to formate in the wild-type strain. That's the reason why formate was also tested for its inductive effect on *cgl0833*.

5. The CRISPR data was well presented in the paper. It may be interesting if the authors try to directly knockout the genes that were in question. Specifically, as they stated lower expression of Cgl0833 may be beneficial, it will be interesting if they can characterize the result of the gene being entirely knocked out.

Response:

Thanks for your helpful comments. We have constructed *cgl0653*- and *cgl0833*-deleted mutants and test their tolerance to methanol. The results have been added in the revised manuscript as Fig. 5f (*cgl0653* deletion) and Fig. 6f (*cgl0833*

deletion). As expected, deletion of *cgl0653* or *cgl0833* in wild-type *C. glutamicum* improved cellular tolerance to methanol, which was consistent with the gene knock-down results.

Please refer to P15, L315: “We further knocked out *cgl0653* in the wild-type *C. glutamicum* ATCC 13032 and observed a similar growth advantage in the presence of methanol stress (Fig. 5f).”

Please refer to Fig. 5f:

f

Fig. 5 f Effects of *cgl0653* knock-out on methanol tolerance. CGXII minimal medium supplemented with 5 g/L glucose and 30 g/L methanol was used to cultivate *C. glutamicum* ATCC 13032 and derivatives. Error bars indicate standard deviations from three parallel experiments.

Please refer to P17, L357: “Knock-out of *cgl0833* led to a similar growth advantage in the presence of methanol stress (Fig. 6f).”

Please refer to Fig. 6f:

f

Fig. 6 f Effects of *cgl0833* knock-out on methanol tolerance. CGXII minimal medium supplemented with 5 g/L glucose and 30 g/L methanol was used to cultivate *C. glutamicum* ATCC 13032 and derivatives. Error bars indicate standard deviations from three parallel experiments.

Reviewers' comments:

Reviewer #2 (Remarks to the Author):

The author responded to all comments, and now the manuscript will be suitable for publication.

Reviewer #3 (Remarks to the Author):

The author added a ^{13}C experiment to demonstrate methanol incorporation. However, other major points remain unchanged.

The authors claimed that this is a new strategy by increasing tolerance to improve production. In fact, this is a common approach, and companies are routinely using this technique to improve production. There is really nothing new regarding the strategy. Furthermore, this is also a follow up from their previous publication (*Metab. Eng.* 49, 220–231 (2018)). Their particular implementation may be of interest to specialized groups, but not to the general audience.

Another point of concern is the excessively high methanol:xylose consumption ratio (greater than 7:1). They cited their previous paper (*Metab. Eng.* 49, 220–231 (2018)) for the similar observation. However, this is inconsistent with the pathway shown in the manuscript. Either the measurement is inaccurate or the pathway is incomplete. Since methanol is highly volatile and they conducted the fermentation under aerobic shaking, it is possible that there is systematic measurement error involved. The other paper they cited is using a different pathway, and thus the argument is invalid.

Responses to the reviewers' comments point-by-point

Thank you for the reviewers' comments, which are very helpful for us to improve our manuscript. We carefully considered all the comments and revised the manuscript accordingly. A marked-up manuscript with the changes highlighted in red have been uploaded. We hope that the answers and the changes in the manuscript will enable our manuscript to win your satisfaction.

Reviewer #2 (Remarks to the Author):

The author responded to all comments, and now the manuscript will be suitable for publication.

Response:

Thanks for your comments. We are glad that our revised manuscript wins your satisfaction.

Reviewer #3 (Remarks to the Author):

The author added a 13C experiment to demonstrate methanol incorporation. However, other major points remain unchanged.

The authors claimed that this is a new strategy by increasing tolerance to improve production. In fact, this is a common approach, and companies are routinely using this technique to improve production. There is really nothing new regarding the strategy. Furthermore, this is also a follow up from their previous publication (Metab. Eng. 49, 220–231 (2018)). Their particular implementation may be of interest to specialized groups, but not to the general audience.

Response:

Thanks for your helpful comments. According to your and the editor's suggestions, we have toned down the statement in abstract that tolerance engineering is a new strategy. We also went through the manuscript to avoid such statement.

Please refer to P2, L23, Abstract: “Herein, we provide a new strategy to enhance methanol bioconversion of synthetic methylotrophs by improving cellular tolerance to methanol.” was changed to “Herein, we enhanced methanol bioconversion of synthetic methylotrophs by improving cellular tolerance to methanol.”.

Another point of concern is the excessively high methanol:xylose consumption ratio

(greater than 7:1). They cited their previous paper (*Metab. Eng.* 49, 220–231 (2018)) for the similar observation. However, this is inconsistent with the pathway shown in the manuscript. Either the measurement is inaccurate or the pathway is incomplete. Since methanol is highly volatile and they conducted the fermentation under aerobic shaking, it is possible that there is systematic measurement error involved. The other paper they cited is using a different pathway, and thus the argument is invalid.

Response:

Thanks for your helpful comments. We would like to explain the rationality of the observed high methanol:xylose consumption ratio from two aspects, **our data/analyses and the aforementioned paper published by another group** (*Nat. Commun.* 2018, 9, 1508).

If all the formaldehyde acceptor Ru5P is totally produced from xylose metabolism, the *rpiB* auxotroph should uptake methanol plus xylose in a 1:1 molar ratio. **However, Ru5P can also be regenerated from the engineered RuMP cycle or oxidative pentose phosphate pathway (PPP). In this case, another molecular of methanol is incorporated with Ru5P, resulting in higher methanol:xylose consumption ratio than 1:1.** **First**, in the previous Fig. 1, we did not show the oxidative PPP, which may cause confusion. **We have added the oxidative PPP in the revised Fig. 1.** **Second**, we performed an *in-silico* analysis to demonstrate the rationality of methanol:xylose consumption at a molar ratio of 7:1, using a *C. glutamicum* genome-scale metabolic model *iCW773* with methanol utilization enzymes (Mdh, Hps, and Phi) added and RpiB deleted. We know the *in-silico* analysis cannot 100 per cent simulate the actual cellular metabolism in the engineered strain, whereas it can simulate a possible metabolic flux distribution when methanol and xylose are consumed at a ratio of 7:1. **According to the *in-silico* analysis (shown in Fig. R1, the original data file was also uploaded), methanol can be consumed with xylose at a ratio of 7:1. As expected, a significant amount of fructose-6-phosphate flux goes to RuMP cycle and oxidative PPP to regenerate Ru5P, which serves as an acceptor to assimilate another molecular of formaldehyde generated by methanol oxidation.** **Third, several completely ¹³C-labeled amino acids were detected in biomass.** Taking L-alanine as an example, it is produced with pyruvate as a precursor. If one molecular ¹³C-methanol (●) is assimilated with one molecular non-labeled xylose (○○○○), half amount of pyruvate should be non-labeled (○○○) and another half should be labeled with only one ¹³C (●○○), so does L-alanine. However, our data shows that **48.90% of**

L-alanine is non-labeled, 30.22% is M+1, 15.66% is M+2, and 5.22% is M+3 (completely ^{13}C -labeled) (Fig. 2e), suggesting methanol and xylose are consumed at a ratio much higher than 1:1. **Fourth**, as mentioned in methods section, to avoid evaporation of methanol, the shake flasks were covered with a sealing membrane and evaporation controls were conducted. By this means, methanol evaporation can be largely decreased (Fig. 2c) and methanol consumption can be accurately determined.

All the data can explain the rationality of the observed high methanol:xylose consumption ratio.

The revised Fig. 1:

Fig. 1 Improving the tolerance to methanol via ALE. **a** Detailed enzymatic reactions and metabolic pathways of the methanol-dependent *C. glutamicum*. Enzymes: methanol dehydrogenase (Mdh), 3-hexulose-6-phosphate synthase (Hps), 6-phospho-3-hexuloisomerase (Phi), mycothiol-dependent formaldehyde dehydrogenase (AdhE), acetaldehyde dehydrogenase (Ald), xylose isomerase (XylA), xylulokinase (XylB), ribose phosphate isomerase (RpiB), ribulose phosphate epimerase (Rpe), transketolase (Tkt), transaldolase (Tal), **glucose-6-phosphate isomerase; (Pgi), glucose-6-phosphate 1-dehydrogenase (Zwf), 6-phosphogluconolactonase (Pgl), 6-phosphogluconate dehydrogenase (Gnd).** Metabolites: ribose-5-phosphate (R5P), ribulose-5-phosphate (Ru5P),

xylulose-5-phosphate (Xu5P), glyceraldehyde-3-phosphate (G3P), erythrose-4-phosphate (E4P), sedoheptulose-7-phosphate (S7P), fructose-6-phosphate (F6P), hexulose-6-phosphate (H6P), **glucose-6-phosphate (G6P), 6-phospho-glucono-1,5-lactone (6PGL), 6-phospho-gluconate (6PG).**

Fig. R1 for point-by-point response:

Fig. R1 *In-silico* simulation of methanol and xylose co-utilization with a ratio of 7:1. The *C. glutamicum* genome-scale metabolic model *iCW773* (Zhang et al. Biotechnol. Biofuels 2017, 10, 169) with methanol utilization enzymes (Mdh, Hps, and Phi) added and RpiB deleted was used for *in-silico* analysis. Methanol and xylose inputs were set at 7 mmol/gCDW·h and 1 mmol/gCDW·h, respectively, to simulate a co-utilization ratio of 7:1. The red numbers in parentheses represent metabolic fluxes of reactions. Partial metabolic reactions are plotted here. The full data set (7methanol-1xylose.txt) was also uploaded.

Next, we would like to explain the reason why this reference published by Meyer et al. (Nat. Commun. 2018, 9, 1508) can support our observation of methanol:xylose consumption at a ratio higher than 1:1. **First**, in this reference, the authors used gluconate as a co-substrate for methanol assimilation, which is different from our study that used xylose as a co-substrate. However, in both engineering strategies, engineered gluconate and xylose catabolic pathways led to generation of Ru5P and condensation of formaldehyde and Ru5P was the only active pathway for generation of F6P (Fig. 2 of the reference is shown below). **Therefore**, despite the differences in specific pathway engineering strategies, the core

concepts of two studies are the same, which are providing formaldehyde acceptor Ru5P from a co-substrate (xylose or gluconate), and meanwhile blocking all the Ru5P catabolism pathways (deleting *rpiB*) except the condensation of formaldehyde and Ru5P into F6P that support cell growth (as reviewed in Wang et al. Trends Biotechnol. 2020, 10.1016/j.tibtech.2019.12.013). **Second**, as shown in Supplementary Fig. 5 of this reference (shown below), **much more methanol was utilized than gluconate (Please notice the y-axes of channels a and b. Approximately 35 mM methanol and 2 mM gluconate were consumed).**

Fig. 2 from the reference:

Fig. 2 Metabolism of a methanol-essential *Escherichia coli* strain. The introduced synthetic pathway is depicted in orange and comprises methanol dehydrogenase (Mdh), 3-hexulose-6-phosphate synthase (Hps), and 6-phospho-3-hexuloisomerase (Phi). Identified knockouts leading to methanol essentiality in the case of gluconate as a carbon source are shown in red; phosphogluconate dehydratase (*edf*) and ribose-5-phosphate isomerase (*rpiAB*). The knockout for reducing TCA cycle activity and mimicking natural methylotrophs is shown in blue; here malate dehydrogenase (*maldh*). Names and abbreviations are given according to the Biocyc database (<http://www.biocyc.org>)

Supplementary Fig. 5 from the reference:

Supplementary Figure 5 Methanol and gluconate consumption of MeSV2.2 (black circles) and wild type *E. coli* containing empty vector control (open squares); n=3. Data presented as mean \pm standard deviation. a) Methanol consumption b) gluconate consumption c) growth during consumption.

REVIEWERS' COMMENTS:

Reviewer #4 (Remarks to the Author):

The reviewer considers the explanation for high methanol:xylose consumption is acceptable. The anaplerotic reaction in the cell can balance the metabolism that may lead to this high ratio. The authors perform an in-silico analysis based on a genome-scale metabolic model to describe the anaplerotic reaction and explain the changes. They also analyze ¹³C-labeled amino acids to calculate the metabolic flux. Actually, the adaptive laboratory evolution is usually a stress-based strategy to regulate or even change the cell metabolism, which can then influence the nutrient absorption.

The adaptive laboratory evolution is an important approach adopted in this article, however this approach is not mentioned even in the Introduction. Therefore, the reviewer suggests to add a brief introduction and highlight that in title, such as "Adaptive laboratory evolution enhances methanol conversion in engineered *Corynebacterium glutamicum*".

Reviewer #4 (Remarks to the Author):

The reviewer considers the explanation for high methanol:xylose consumption is acceptable. The anaplerotic reaction in the cell can balance the metabolism that may lead to this high ratio. The authors perform an in-silico analysis based on a genome-scale metabolic model to describe the anaplerotic reaction and explain the changes. They also analyze 13C-labeled amino acids to calculate the metabolic flux. Actually, the adaptive laboratory evolution is usually a stress-based strategy to regulate or even change the cell metabolism, which can then influence the nutrient absorption.

*The adaptive laboratory evolution is an important approach adopted in this article, however this approach is not mentioned even in the Introduction. Therefore, the reviewer suggests to add a brief introduction and highlight that in title, such as “Adaptive laboratory evolution enhances methanol conversion in engineered *Corynebacterium glutamicum*”.*

Response:

Thanks for your helpful comments. We have added a brief introduction to adaptive laboratory evolution in the Introduction and revised the title to highlight the adaptive laboratory evolution.

Please refer to the revised title: “Adaptive laboratory evolution enhances methanol tolerance and conversion in engineered *Corynebacterium glutamicum*”.

Please refer to P4, L74: “Adaptive laboratory evolution (ALE) strategies, which allow occurrence and selection of beneficial mutations in an unbiased fashion²⁵, were then applied to effectively improve cell growth on methanol and co-substrates²⁶⁻²⁸.”